# Δ-DiT: Accelerating Diffusion Transformers without training via Denoising Property Alignment

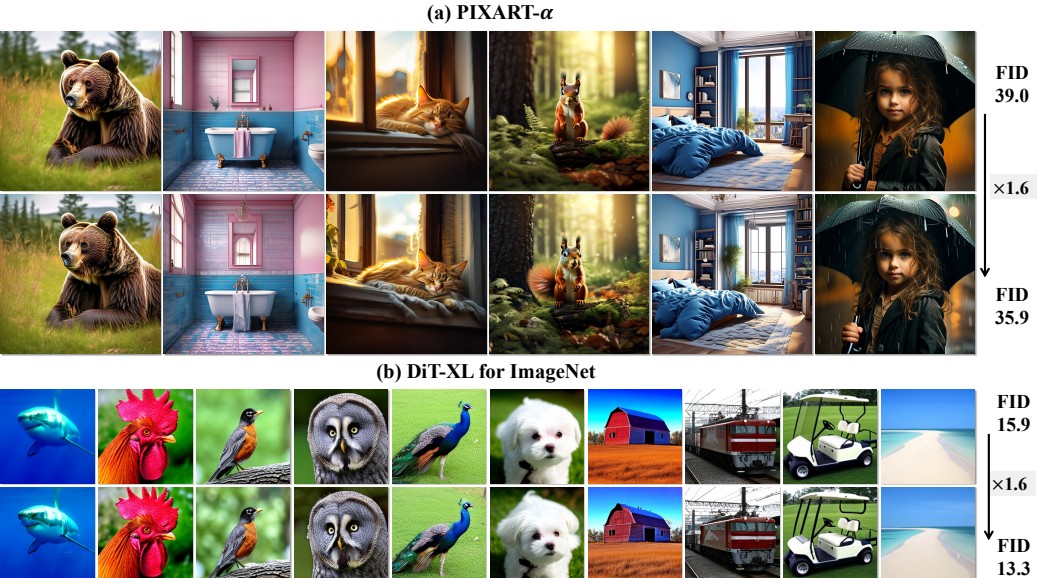

(a) PIXART-α

(b) DiT-XL for ImageNet

Figure 1: Accelerating PIXART-α and DiT-XL by $1.6\times$ speedup with 20 DPMSolver++ steps.

## Abstract

Diffusion models are now commonly used for producing high-quality and diverse images, but the iterative denoising process is time-intensive, limiting their usage in real-time applications. As a result, various acceleration techniques have been developed, though these primarily target UNet-based architectures and are not directly applicable to Transformer-based diffusion models (DiT). To address the specific challenges of the DiT architecture, we first analyze the relationship between the depth of DiT blocks and the quality of image generation. While skipping blocks can lead to large degradations in generation quality, we propose the Δ-Cache method, which captures and stores the incremental changes of different blocks, thereby mitigating the performance gap and maintaining closer alignment with the original results. Our analysis indicates that the shallow DiT blocks primarily define the global structure of images such as compositions and outlines, while the deep blocks mainly refine details, and the role of middle blocks lies between the two. Based on this, we introduce a denoising property alignment method that selectively bypasses computations of different blocks at various timesteps while preserving performance. Comprehensive experiments on PIXART-α and DiT-XL demonstrate that Δ-DiT achieves a $1.6\times$ speedup in 20-step generation and enhances performance in most cases. In the 4-step consistent model generation scenario, and with a more demanding $1.12\times$ acceleration, our approach significantly outperforms existing methods.

# 1 INTRODUCTION

In recent years, the field of generative models has experienced rapid advancements. Among these, diffusion models (Ho et al., 2020; Rombach et al., 2022; Song et al., 2021b) have emerged as pivotal, attracting widespread attention for their ability to generate high-quality and diverse images (Dhariwal & Nichol, 2021). This has also spurred the development of many meaningful applications, such as image editing (Kawar et al., 2023; Zhang et al., 2023a), 3D generation (Tang et al., 2023b;a; Mo et al., 2023), and video generation (Wu et al., 2023a; Khachatryan et al., 2023; Blattmann et al., 2023; Luo et al., 2023b). Although diffusion models have strong generation capabilities, their iterative denoising nature results in poor real-time performance. Subsequently, numerous inference acceleration frameworks have been proposed, which include general model compression methods for denosing network (Fang et al., 2023; Zhang et al., 2024a; Shang et al., 2023; So et al., 2023; Kim et al., 2023; Salimans & Ho, 2022; Luo et al., 2023a; Zhao et al., 2023; Sauer et al., 2023), fast sampling solver (Song et al., 2021a; Lu et al., 2022a;b; Zhang & Chen, 2023; Karras et al., 2022), and cache-based acceleration methods (Ma et al., 2023; Li et al., 2023b). However, almost all of these acceleration techniques are designed for the UNet-based (Ronneberger et al., 2015) architecture.

Recently, Diffusion Transformers (DiT) (Peebles & Xie, 2023) have emerged as dominant foundational models, exemplified by PIXART-$\alpha$ (Chen et al., 2023), SD3.0 (Esser et al., 2024), and Sora (Brooks et al., 2024). Despite this success, the acceleration of DiT inference is under-explored. Existing methods, such as early stopping (Moon et al., 2023), require retraining and are unsuitable for small-step generation. DiT's isotropic architecture, with no long skip connections found in UNet, makes it difficult to apply UNet-based acceleration techniques. For instance, cache-based methods (Ma et al., 2023; Li et al., 2023b) may result in information loss, as DiT lacks the long shortcuts that facilitate feature reuse in UNet. Moreover, skipping computations for branches can introduce significant degradations. To address this, we propose $\Delta$-**Cache**, a caching method tailored for transformer architectures that caches $\Delta$ change between different blocks instead of the original feature maps, preventing large degradation and making caching more effective for DiT.

In our caching framework, we first investigate the degradations introduced by caching at different blocks within the transformer. We observe that the shallow transformer blocks in DiT primarily define the global structure of images such as compositions, and outlines, while the deep blocks focus on refining image details, as illustrated in Figure 2. While previous studies (Wang & Vastola, 2023; Liu et al., 2023; Hertz et al., 2023) have pointed out a property of diffusion models: creating contours in the earlier timesteps (early denoising stage) and generating details in the later timesteps (later stage). Building on this property and our findings, this paper proposes a denoising property alignment inference acceleration method, $\Delta$-DiT. Specifically, the method applies $\Delta$-Cache to the deeper blocks during the early denoising stage to soften the details and preserve the contours, while applying $\Delta$-Cache to the shallow blocks during later sampling to maintain the details, thus aligning with the property of the diffusion models (Wang & Vastola, 2023; Liu et al., 2023; Hertz et al., 2023). We evaluated our approach across multiple datasets, including MS-

**Prompt:** *a couple of fire trucks that are by a motorcycle*

(a) No $\Delta$-Cache

(b) $\Delta$-Cache shallow blocks

(c) $\Delta$-Cache middle blocks

(d) $\Delta$-Cache deep blocks

Figure 2: Images generated by $\Delta$-Cache for various blocks within the DiT.

COCO2017 (Lin et al., 2014) and PartiPrompts (Yu et al., 2022), using various DiT architectures such as PIXART-$\alpha$ (Chen et al., 2023), DiT-XL (Peebles & Xie, 2023), and PIXART-$\alpha$-LCM (Chen et al., 2023; Luo et al., 2023a). Extensive quantitative results confirm the effectiveness of our method. In the 20-step generation, we achieved a 1.6x speedup, with FID improving from 39.002 to 35.882. In the more challenging 4-step generation scenarios, our method also significantly outperformed existing baselines in terms of FID score from 44.198 to 40.118. The contributions of our paper are three-fold:

- We adapt the caching method to transformers using $\Delta$-Cache, which stores the incremental changes in feature maps. Furthermore, we identify a correlation between different DiT

blocks and the final generation results: shallow blocks focus on generating image outlines, while deep blocks emphasize image details.

- To align with the denoising property of generating outlines first and details later, we propose a training-free acceleration framework, termed $\Delta$-DiT. Specifically, we accelerates inference by caching deep blocks during the early stages of denoising and shallower blocks in the later stages.

- We show empirically that $\Delta$-DiT achieves a 1.6× speedup in 20-step generation while improving image quality. On more challenging image generation scenarios (eg. 4-step), $\Delta$-DiT also outperforms existing approaches in generation quality by a significant margin.

## 2  RELATED WORK

**Efficient Diffusion Model.** To improve the real-time performance of diffusion models, various lightweight and acceleration techniques have emerged. Currently, methods for accelerating diffusion models for image generation can be broadly categorized into three perspectives: a lightweight denoising model, and reduced denoising timestep, and the intersection of the model and timestep dimension. Similar to traditional model compression, many efforts focus on pruning (Fang et al., 2023; Zhang et al., 2024a), quantization (Shang et al., 2023; So et al., 2023; He et al., 2023; Li et al., 2023c), and distillation (Kim et al., 2023; Salimans & Ho, 2022; Luo et al., 2023a; Zhao et al., 2023; Sauer et al., 2023) to obtain a smaller yet comparable denoising network. Besides, reduced denoising timestep is a unique dimension for diffusion models. Most methods currently focus on exploring efficient ODE solvers (Song et al., 2021a; Lu et al., 2022a;b; Zhang & Chen, 2023; Karras et al., 2022), aiming to obtain high-quality images with fewer sampling steps. LCM (Song et al., 2023; Luo et al., 2023a) proposes consistency loss and knowledge distillation to achieve the goal of fewer steps. Lastly, there's a focus on jointly optimizing denoising modes and timesteps. For instance, OMS-DPM (Liu et al., 2023) and Autodiffusion (Li et al., 2023a) simultaneously optimize skips and allocate noise estimation networks of specific sizes for each timestep. However, most of the aforementioned work is implemented and validated on the UNet architecture. One previous work (Moon et al., 2023) proposes an early stopping strategy for DiT, which cannot be easily transferred to fewer timestep settings. Therefore, there is currently a lack of novel acceleration methods specifically designed for the DiT architecture.

**Cache Mechanism.** The cache mechanism is a key concept in computer systems, designed to temporarily store data for reuse, improving processing efficiency. In Large Language Models, the KV cache (Zhang et al., 2023b; Ge et al., 2023) is widely used, caching key and value matrices from attention blocks to accelerate inference. Cache-based techniques have also been applied to diffusion models, with DeepCache (Ma et al., 2023) accelerating UNet by caching feature maps from up-sampling blocks, and Faster Diffusion (Li et al., 2023b) optimizing computation by caching outputs from UNet encoders. On a finer granularity, studies such as (Zhang et al., 2024b; Wimbauer et al., 2024; So et al., 2024) focus on caching feature maps within specific blocks to save computations. These methods target feature maps at various locations and in differing quantities as caching objectives, and most of them are targeted at the UNet architecture. However, this paper introduces a feature map offsets caching method, specifically tailored to the isotropic architecture of DiT.

## 3  PRELIMINARY

The concept of diffusion originates from a branch of non-equilibrium thermodynamics (De Groot & Mazur, 2013) in physics. In recent years, researchers have applied this concept to image generation (Ho et al., 2020; Rombach et al., 2022; Song et al., 2021b; Dhariwal & Nichol, 2021; Song & Ermon, 2019), transforming the process into two stages: noise diffusion and denoising.

**Noising Process.** This is also the training phase of the diffusion model. Given an original image $x_0$ and a random time step $t \in [1, T]$ (where $T$ is the total steps), the image after $t$ steps of diffusion is $\sqrt{\bar{\alpha}_t} x_0 + \sqrt{1 - \bar{\alpha}_t} \epsilon$, where $\bar{\alpha}_t$ is constant related to $t$. The noise estimation network is then used to estimate the noise in the diffused result, making the estimated noise $\epsilon_\theta$ as close as possible to the actual noise $\epsilon$ added during diffusion. The learning objective is defined as follows (Ho et al., 2020):

$$\mathcal{L}(\boldsymbol{\theta}) = \mathbb{E}_{t, \boldsymbol{x_0} \sim q(\boldsymbol{x}), \boldsymbol{\epsilon} \sim \mathcal{N}(0,1)} \left[ \| \boldsymbol{\epsilon} - \boldsymbol{\epsilon_\theta}(\sqrt{\bar{\alpha}_t} \boldsymbol{x_0} + \sqrt{1 - \bar{\alpha}_t} \boldsymbol{\epsilon}, t) \|^2 \right], \tag{1}$$

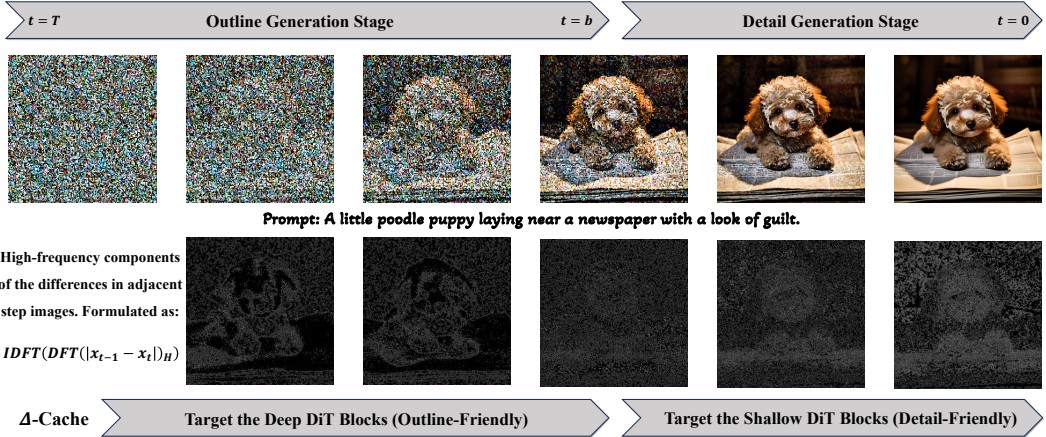

Figure 3: **Overview of the $\Delta$-DiT**: The denoising property emphasizes generating outlines early in denoising and details later. Our previously proposed $\Delta$-Cache method caches deep blocks for outline-friendly generation and shallow blocks for detail-friendly results. In the $\Delta$-DiT, the properties of Denoising and $\Delta$-Cache are aligned in stages, that is, $\Delta$-Cache is applied to the deep blocks in the DiT during the early outline generation stage of the diffusion model, and on shallow blocks during the detail generation stage. The stage is bounded by a hyperparameter $b$.

where $q(\boldsymbol{x})$ is the dataset distribution, and $\mathcal{N}$ is the Gaussian distribution. In most current works, the noise estimation networks are mostly based on UNet architecture. However, in isotropic architectures like DiT, $\boldsymbol{\epsilon_\theta}(\boldsymbol{x}_t)$ can be further transformed into $\boldsymbol{f}_{N_b}^t(\boldsymbol{f}_{N_b-1}^t(\cdots(\boldsymbol{f}_1^t(\boldsymbol{x}_t)))) = \boldsymbol{f}_{N_b}^t \circ \boldsymbol{f}_{N_b-1}^t \circ \cdots \circ \boldsymbol{f}_1^t(\boldsymbol{x}_t) = \boldsymbol{F}_{1:N_b}^t(\boldsymbol{x}_t)$, where $\boldsymbol{f}_n^t$ represents the mapping of the $n$-th DiT block at timestep $t$, and $\boldsymbol{F}_{1:N_b}^t$ represents the mapping of the first to the $N_b$-th DiT blocks. $N_b$ denotes the number of blocks.

**Denoising Process.** During this process, Gaussian noise is iteratively denoised into a generated image, and our goal is to accelerate this denoising process without requiring additional training. Initially, a random Gaussian noise $\boldsymbol{x}_T$ is given. It is then fed into the denoising network $\boldsymbol{\epsilon_\theta}$ to obtain the estimated noise $\boldsymbol{\epsilon_\theta}(\boldsymbol{x}_T)$. With sampling solvers, the noisy image is denoised to produce the denoised sample $\boldsymbol{x}_{T-1}$ for each timestep. After iterating this process $T$ times, the final generated image is obtained. Using the DDPM (Ho et al., 2020) solver as an example, the iterative denoising process can be defined as follows:

$$\boldsymbol{x}_{t-1} = \frac{1}{\sqrt{\alpha_t}}\left(\boldsymbol{x}_t - \frac{\beta_t}{\sqrt{1-\bar{\alpha}_t}}\boldsymbol{\epsilon_\theta}(\boldsymbol{x}_t, t)\right) + \sigma_t \boldsymbol{z}, \tag{2}$$

where $\alpha_t$, $\beta_t$ and $\sigma_t$ is constant related to $t$, and $\boldsymbol{z} \sim \mathcal{N}(\boldsymbol{0}, \boldsymbol{I})$. For other solvers (Song et al., 2021a; Lu et al., 2022a;b; Zhang & Chen, 2023; Karras et al., 2022), the sampling formula differs slightly from Eq. 2, but they are all functions of $\boldsymbol{x}_t$ and $\boldsymbol{\epsilon_\theta}$. In many scenarios, the noise estimation network $\boldsymbol{\epsilon_\theta}(\boldsymbol{x}_t, t, \boldsymbol{c})$ has another input $\boldsymbol{c}$. It is conditional control information, which can be either a class embedding or a text embedding.

## 4 METHODOLOGY

In this section, we present our denoising property alignment method for training-free acceleration of DiT. First, we introduce $\Delta$-Cache, a novel caching method specifically designed for DiT. Then, leveraging this framework, we explore the specific effects of different parts of blocks on generation. Finally, by integrating the previous findings with the properties of the denoising process, we propose $\Delta$-DiT to accelerate DiT generation. The overall framework is shown in Figure 3.

### 4.1 EFFECT OF DiT BLOCKS ON GENERATION

In accelerating DiT, methods like skipping blocks (Raposo et al., 2024) offer a straightforward way to reduce computational overhead. However, skipping blocks during inference without additional training introduces significant degradations from the original results as shown in Figure 4a. Each DiT

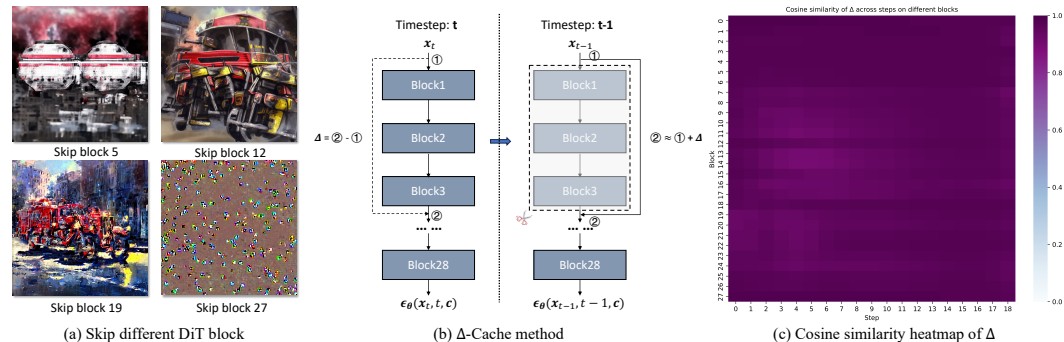

(a) Skip different DiT block   (b) Δ-Cache method   (c) Cosine similarity heatmap of Δ

Figure 4: (a) **Visualization of images generated by skipping different blocks**. (b) **Illustration of Δ-Cache**. The difference between the feature maps at both ends of the block is used as Δ. Then, Δ is employed in the next step to compensate for the skipped computation of the block. (c) **Cosine similarity heatmap of Δ**. Similarity of Δ of blocks with different steps and different depths.

block plays a critical role in estimating noise at each timestep, and directly omitting critical blocks can cause the final image to deteriorate into noise. Thus, it is necessary to compensate for the large discrepancies introduced by skipping blocks to maintain image quality.

**Δ-Cache**. Inspired by recent cache-based methods (Ma et al., 2023; Li et al., 2023b), caching and reusing previous feature maps offers a potential solution. However, this approach cannot be directly applied to transformer architectures due to the absence of long skip-connection. Therefore, we propose Δ-Cache to cache incremental changes between blocks (e.g., in Figure 4b, the differences between the features at points ① and ② is cached), which based on Δ exhibits a high degree of similarity. As shown in Figure 4c, the cosine similarity of Δ between blocks at the same position across adjacent timesteps remains above 0.9. This high similarity allows the Δ-Cache method to incur minimal information loss. During denoising, some timesteps skip the computation of blocks and apply the cached Δ as compensation to minimize the degradation in the inference. Based on the mathematical framework described in Section 3, Δ-Cache process can be defined as follows:

$$
\begin{aligned}
\boldsymbol{x}_{t-1} &= \frac{1}{\sqrt{\alpha_t}} \left( \boldsymbol{x}_t - \frac{\beta_t}{\sqrt{1-\bar{\alpha}_t}} \boldsymbol{\epsilon}_\theta(\boldsymbol{x}_t, t) \right) + \sigma_t \boldsymbol{z} = \frac{1}{\sqrt{\alpha_t}} \left( \boldsymbol{x}_t - \frac{\beta_t}{\sqrt{1-\bar{\alpha}_t}} F_{1:N_b}^t(\boldsymbol{x}_t) \right) + \sigma_t \boldsymbol{z} \\
&= \frac{1}{\sqrt{\alpha_t}} \left( \boldsymbol{x}_t - \frac{\beta_t}{\sqrt{1-\bar{\alpha}_t}} F_{I+N_c:N_b}^t \big( F_{1:I+N_c}^t(\boldsymbol{x}_t) \big) \right) + \sigma_t \boldsymbol{z} \\
&= \frac{1}{\sqrt{\alpha_t}} \left( \boldsymbol{x}_t - \frac{\beta_t}{\sqrt{1-\bar{\alpha}_t}} F_{I+N_c:N_b}^t \big( F_{1:I}^t(\boldsymbol{x}_t) + \underline{F_{1:I+N_c}^t(\boldsymbol{x}_t) - F_{1:I}^t(\boldsymbol{x}_t)} \big) \right) + \sigma_t \boldsymbol{z} \\
&\approx \frac{1}{\sqrt{\alpha_t}} \left( \boldsymbol{x}_t - \frac{\beta_t}{\sqrt{1-\bar{\alpha}_t}} F_{I+N_c:N_b}^t \big( F_{1:I}^t(\boldsymbol{x}_t) + \underline{F_{1:I+N_c}^{t+1}(\boldsymbol{x}_{t+1}) - F_{1:I}^{t+1}(\boldsymbol{x}_{t+1})} \big) \right) + \sigma_t \boldsymbol{z}
\end{aligned}
\tag{3}
$$

Here, the underlined part is Δ, $I$ indicates the starting block position of the Δ-Cache, $N_b$ denotes the number of blocks and $N_c$ refers to the number of cached blocks.

**Qualitative Analysis**. Within the caching framework, we aim to explore the impact of different blocks on the final generated image. Given that the effect of Δ-Cache on individual blocks is minimal (as shown in Figure 4c), we analyze the impact at a coarser granularity to make the results more discernible, rather than performing a block-by-block analysis. For a 28-block transformer like DiT-XL and PIXART-$\alpha$, we divide the network into three main sections: (1) **Shallow Blocks (1-21)**: the first 21 blocks, (2) **Middle Blocks (4-24)**: the middle 21 blocks, and (3) **Deep Blocks (8-28)**: the last 21 blocks. This segmentation allows us to better assess the influence of different regions within the model. As shown in Figure 2, we can conclude that:

1) **For Shallow Blocks**. Applying Δ-Cache to the shallow blocks results in inaccurate outline generation. As shown in Figure 2a (green arrows), the blue car's outline on the right is clear. However, in Figure 2b, the outline is absent, despite the image is better generated in detail.

2) **For Deep Blocks**. In contrast, applying Δ-Cache to the deep blocks preserves the global outline but reduces detail accuracy. As shown in Figure 2d, the blue car's outline is retained, but some noise appears in the finer details.

3) **For Middle Blocks**. $\Delta$-Cache applied to the middle part provides a compromise.

From this qualitative analysis, we can conclude that the shallow blocks of DiT are more related to outline generation, the deep blocks are more connected to detail generation, and the middle blocks represent a balance between the two.

**Quantitative Analysis**. Despite the qualitative analysis indicating a correlation between DiT blocks and the generated output, we further validate this observation through statistical analysis. To quantify the ability to generate details and outlines, we employ the Fourier transform as an effective method (Broughton & Bryan, 2018). In Fourier analysis, high-frequency components correspond to rapid intensity changes, typically associated with details like textures or edges, while low-frequency components represent the global structure or outline. A higher proportion of high-frequency components suggests better detail generation, while strong outline generation reflects more low-frequency components. Specifically, we can calculate the relative log magnitude from the Fourier transform as follows:

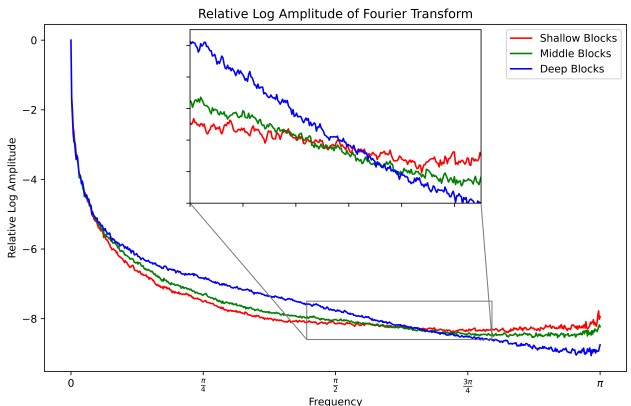

Figure 5: Fourier relative log magnitude of images generated by applying $\Delta$-Cache to various blocks of the DiT.

$$\mathcal{F}(\boldsymbol{x}) = \mathcal{F}(u, v) = \sum_{x=1}^{H} \sum_{y=1}^{W} \boldsymbol{x}(x, y) \cdot e^{-2\pi i \left( \frac{u}{H} x + \frac{v}{W} y \right)} \tag{4}$$

$$\text{Relative Log Magnitude}(u, v) = log\left( \frac{\sqrt{\text{Re}(\mathcal{F}(u,v))^2 + \text{Im}(\mathcal{F}(u,v))^2}}{\max\limits_{u,v} \sqrt{\text{Re}(\mathcal{F}(u,v))^2 + \text{Im}(\mathcal{F}(u,v))^2}} \right) \tag{5}$$

Where $\mathcal{F}$ represents the Fourier transform, $\boldsymbol{x}(x, y)$ denotes the pixel values of the image, and $H$ and $W$ stand for the image's height and width, respectively. The coordinates $(u, v)$ correspond to the frequency domain. Re($\cdot$) and Im($\cdot$) represent the real parts and imaginary parts of a complex number, respectively.

Using Fourier analysis, we can further compare how different parts of the blocks influence the generated image. To draw statistical conclusions, we perform this analysis on a subset of the MS-COCO2017 dataset Zhao et al. (2024). We compute the relative log magnitude of the Fourier transform for images generated under three different settings, converting it into a radial relative magnitude-frequency space, and calculated the expectation across the dataset. The results are presented in Figure 5 , leading to the following conclusions:

1) **For Shallow Blocks**. When $\Delta$-Cache is applied to the shallow blocks, where the shallow blocks are lossy, the generated images show a high proportion of high-frequency components, indicating strong detail generation, that is, friendly to detail generation.

2) **For Deep Blocks**. Applying $\Delta$-Cache to deep blocks exhibits a high proportion of low-frequency components, indicating strong outline generation, that is, friendly to outline generation.

3) **For Middle Blocks**. Middle blocks with $\Delta$-Cache applied exhibit a balance between these two extremes.

These findings align with the qualitative analysis presented earlier.

### 4.2 ACCELERATING DiT VIA DENOISING PROPERTY ALIGNMENT

**Denoising property**. Previous research (Wang & Vastola, 2023; Liu et al., 2023; Hertz et al., 2023) has demonstrated that the denoising process in diffusion models follows a generation pattern. During

the early stages, these models primarily generate image outlines, while the focus shifts to details in the later stages. To further illustrate this, the bottom part of Figure 3 shows the difference between images generated at adjacent timesteps. After applying a Fourier transform to this difference and extracting the high-frequency components—representing areas with significant changes. We can observe that in the early denoising stages (first two images in Figure 3), these regions primarily capture the outline of the dog, while in later stages (last two images), the focus shifts to finer details like the dog's fur.

**Denoising property alignment framework**. Therefore, in terms of outline and detail generation, we can align the $\Delta$-Cache with the denoising property. As shown in Figure 3, image denoising at various timesteps (from $t = T$ to $t = 0$) is presented. Since denoising properties emphasize outline generation in the early stages, and the deeper blocks in the $\Delta$-Cache method are more suited for generating outlines, $\Delta$-Cache is applied to the deep blocks during the outline generation stage. Conversely, as diffusion models focus more on detail generation in the later stages, and the shallow blocks of the $\Delta$-Cache method are more appropriate for detail generation, $\Delta$-Cache is applied to the shallow blocks during the detail generation stage.

To this end, we propose a training-free framework termed $\Delta$-DiT, which can generate images with better quality. In our framework, we introduce two hyperparameters. One is denoted as $b$, representing the boundary between the outline generation stage and the detail generation stage. When $t \leq b$, $\Delta$-Cache is applied to the deep blocks; when $t > b$, $\Delta$-Cache is applied to the shallow blocks. The number of blocks requiring $\Delta$-Cache is determined based on the actual computational requirements. Assuming the computation cost of one block is $M_b$ and the expected total computation cost is $M_g$, as previously mentioned, the cache interval is $N$, and the number of DiT blocks is $N_b$. First, we roughly determine the value of $N$ as:

$$N = \lceil \frac{T \times N_b \times M_b}{M_g} \rceil,\tag{6}$$

In some current low-step scenarios, the value of $N$ is set to 2. After determining $N$, the actual number of blocks to cache at the timestep is:

$$N_c = [(\underbrace{\frac{M_g - (T \bmod N) \times N_b \times M_b}{\lfloor T/N \rfloor \times M_b}}_{\text{the computation in each } N \text{ step}} - \underbrace{N_b \times M_b}_{\text{the first full DiT}}) / \underbrace{(M_b \times (N-1))}_{\text{the remaining cached steps}}].\tag{7}$$

Once these hyperparameters are determined, the inference process becomes fixed and remains unchanged regardless of the input, enabling acceleration without the need for further training.

## 5 EXPERIMENT

### 5.1 EXPERIMENTAL SETTINGS

**Models, Evaluation Data and Solvers.** We conduct experiments on three diffusion transformer-based architectures: DiT-XL (Peebles & Xie, 2023), PIXART-$\alpha$(Chen et al., 2023), and PIXART-$\alpha$-LCM(Chen et al., 2023; Luo et al., 2023a). For DiT-XL, we generat 50k images using 1000 ImageNet classes (Russakovsky et al., 2015) for evaluation. For the PIXART-$\alpha$ models, we evaluate image quality using 1.632k prompts from PartiPrompt (Yu et al., 2022) and 5k prompts from the MS-COCO2017 validation dataset (Lin et al., 2014). In our main experiment, we use the 20-step DPMSolver++ (Lu et al., 2022b), the default setting for PIXART-$\alpha$. For consistency model generation, we apply the 4-step LCMSolver (Song et al., 2023). To demonstrate the effectiveness of our method, we compare with several fast-generation techniques, including the feature map caching method from Faster Diffusion (Li et al., 2023b), TGATE (Zhang et al., 2024b).

**Evaluation Metrics.** We use a range of metrics to evaluate both generation efficiency and image quality. For generation efficiency, we measure the theoretical computational complexity using MACs and the practical time to generate an image using latency. Lower MACs and latency indicate higher efficiency, while the speedup reflects the acceleration rate. To assess generation quality, we employ widely used metrics such as FID (Heusel et al., 2017), IS (Salimans et al., 2016), and CLIP-Score (Hessel et al., 2021).

Table 1: The MS-COCO2017 and PartiPrompts generation results for PIXART-$\alpha$ are evaluated. Gate is the hyperparameter defined in TGATE (Zhang et al., 2024b). $T$ represents the number of timesteps, and $I$ indicates the starting block index for caching. Latency, measured in milliseconds, is tested on an Nvidia A100 GPU.

| Method | MACs ↓ | Speedup ↑ | Latency ↓ | MS-COCO2017 | | | PartiPrompts |
| | | | | FID ↓ | IS ↑ | CLIP ↑ | CLIP ↑ |
|---|---|---|---|---|---|---|---|
| PIXART-$\alpha$ ($T = 20$) (Chen et al., 2023) | 85.651T | 1.00× | 2290.668 | 39.002 | 31.385 | **30.417** | 30.097 |
| PIXART-$\alpha$ ($T = 13$) (Chen et al., 2023) | 55.673T | 1.54× | 1565.175 | 39.989 | 30.822 | 30.399 | 29.993 |
| Faster Diffusion ($I = 14$) (Li et al., 2023b) | 64.238T | 1.33× | 1777.144 | 41.560 | 31.233 | 30.300 | 29.958 |
| Faster Diffusion ($I = 21$) (Li et al., 2023b) | 53.532T | 1.60× | 1517.698 | 42.763 | 30.316 | 30.227 | 29.922 |
| TGATE (Gate=10) (Zhang et al., 2024b) | 61.075T | 1.40× | 1718.308 | 37.413 | 31.079 | 29.782 | 29.347 |
| TGATE (Gate=8) (Zhang et al., 2024b) | 56.170T | 1.52× | 1603.250 | 37.539 | 30.124 | 29.021 | 28.654 |
| $\Delta$-Cache (Shallow Blocks) | 53.532T | 1.60× | 1522.346 | 41.702 | 30.276 | 30.288 | 29.964 |
| $\Delta$-Cache (Middle Blocks) | 53.532T | 1.60× | 1522.528 | 35.907 | **33.063** | 30.183 | 30.078 |
| $\Delta$-Cache (Deep Blocks) | 53.532T | 1.60× | 1522.669 | **34.819** | 32.736 | 29.898 | 30.099 |
| Ours ($b = 12$) | 53.532T | 1.60× | 1534.551 | 35.882 | 32.222 | 30.404 | **30.123** |

Table 3: The MS-COCO 2017 and PartiPrompts results for the PIXART-$\alpha$-LCM model are evaluated, using the default number of generation steps, $T = 4$.

| Method | MACs ↓ | Speedup ↑ | Latency ↓ | MS-COCO2017 | | | PartiPrompts |
| | | | | FID ↓ | IS ↑ | CLIP ↑ | CLIP ↑ |
|---|---|---|---|---|---|---|---|
| PIXART-$\alpha$-LCM (Chen et al., 2023) | 8.565T | 1.00× | 415.255 | 40.433 | 30.447 | 29.989 | 29.669 |
| Faster Diffusion ($I = 4$) (Li et al., 2023b) | 7.953T | 1.08× | 401.137 | 468.772 | 1.146 | -1.738 | 1.067 |
| Faster Diffusion ($I = 6$) (Li et al., 2023b) | 7.647T | 1.12× | 391.081 | 468.471 | 1.146 | -1.746 | 1.057 |
| TGATE (Gate=2) (Zhang et al., 2024b) | 7.936T | 1.08× | 400.256 | 42.038 | 29.683 | 29.908 | 29.549 |
| TGATE (Gate=1) (Zhang et al., 2024b) | 7.623T | 1.12× | 398.124 | 44.198 | 27.865 | 29.074 | 28.684 |
| Ours ($b = 2, N_c = 4$) | 7.953T | 1.08× | 400.132 | 39.967 | 29.667 | 29.751 | 29.449 |
| Ours ($b = 2, N_c = 6$) | 7.647T | 1.12× | 393.469 | 40.118 | 29.177 | 29.332 | 29.226 |

## 5.2 COMPARISON WITH ACCELERATION METHODS

We comprehensively compare with efficient generation methods for PIXART-$\alpha$ on both the generation efficiency and image quality in Table 1. The proposed method exceeds the baseline PIXART-$\alpha$($T = 20$) on all metrics except for a small gap in the MS-COCO2017 CLIP-Score, with a 1.60× speedup. With similar inference cost, we surpass PIXART-$\alpha$($T = 13$) on all metrics by a large margin (e.g., FID: 39.989 → 35.882). Moreover, our proposed method also outperforms Faster Diffusion and TGATE in all metrics on both datasets with similar or even higher generation efficiency. Finally, to further illustrate the superior generative performance of our method, refer to the visualizations generated by different methods in Figure 8.

In Table 2, we further validate our proposed method on the DiT-XL architecture Peebles & Xie (2023). The method achieves a 1.6× speedup over the baseline DiT-XL ($T = 20$) while improving the FID from 15.893 to 13.289. Additionally, it surpasses Faster Diffusion ($I = 21$) in terms of IS, improving from 416.609 to 442.028 by a significant margin, while maintaining comparable in-

Table 2: Results on the DiT-XL (cfg=4.0). Because the TGATE can only handle cross-attention, it cannot be used for DiT-XL.

| Method | ImageNet-50k | | | |
| | MACs ↓ | Latency ↓ | FID ↓ | IS ↑ |
|---|---|---|---|---|
| DiT-XL ($T = 20$) (Peebles & Xie, 2023) | 4.579T | 578.201 | 15.893 | 440.797 |
| DiT-XL ($T = 13$) (Peebles & Xie, 2023) | 2.976T | 382.607 | 15.982 | 436.730 |
| Faster Diffusion ($I = 14$) (Li et al., 2023b) | 3.434T | 458.409 | 15.084 | 417.903 |
| Faster Diffusion ($I = 21$) (Li et al., 2023b) | 2.862T | 383.812 | 15.145 | 416.609 |
| $\Delta$-Cache (Shallow Blocks) | 2.862T | 367.148 | 15.112 | 420.198 |
| $\Delta$-Cache (Middle Blocks) | 2.862T | 368.984 | 14.270 | **442.921** |
| $\Delta$-Cache (Deep Blocks) | 2.862T | 367.042 | 13.391 | 439.700 |
| Ours ($b = 12$) | 2.862T | 370.290 | **13.289** | 442.028 |

ference speed. Although $\Delta$-Cache does not lead in all metrics, its strong performance in both tables demonstrates its overall effectiveness, offering a favorable balance between quality and efficiency.

Table 4: Performance under different advanced solvers which are measured on MS-COCO2017.

| Solver | PIXART-$\alpha$ | | | + $\Delta$-DiT | | |
|---|---|---|---|---|---|---|
| | FID ↓ | IS ↑ | CLIP ↑ | FID ↓ | IS ↑ | CLIP ↑ |
| EulerD (Karras et al., 2022) | 39.688 | 31.413 | 30.359 | 35.735 | 32.290 | 30.239 |
| DEIS (Zhang & Chen, 2023) | 37.675 | 32.362 | 30.420 | 35.302 | 32.721 | 30.377 |
| DPMSolver++ (Lu et al., 2022a) | 39.002 | 31.385 | 30.417 | 35.882 | 32.222 | 30.404 |

## 5.3 COMPARISON UNDER LCM SETTINGS

Latent Consistency Model (LCM) (Song et al., 2023; Luo et al., 2023a) introduces a method to accelerate the denoising process using consistency loss, reducing timesteps from over 30 to just 4, making it highly difficult to accelerate further. We test our approach in this challenging scenario to assess its generalizability, as shown in Table 3. Methods like Faster Diffusion (Li et al., 2023b), which lack supervision from previous step images, perform poorly in small-step settings, with significantly high FID scores (FID=468.471). While existing approaches like TGATE (Zhang et al., 2024b) achieve reasonable results, they suffer notable performance degradation (FID: 40.433 → 44.198) at an acceleration ratio of approximately 1.12. In contrast, our method maintains superior performance even with better FID scores.

Table 5: Results of opposite denoising property alignment on PIXART-$\alpha$ and DiT-XL.

| Method | FID ↓ | IS ↑ | CLIP ↑ |
|---|---|---|---|
| PIXART-$\alpha$-Ours | 35.882 | 32.222 | 30.404 |
| Opposite | 41.374 | 30.980 | 30.259 |
| DiT-XL-Ours | 13.289 | 442.028 | / |
| Opposite | 15.255 | 426.949 | / |

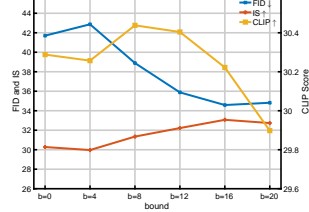

Figure 6: The choice of bound value $b$.

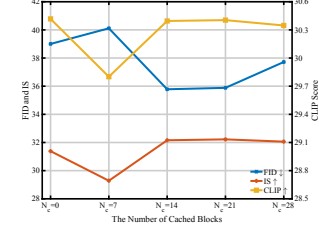

Figure 7: The choice of total cached blocks $N_c$.

## 5.4 ABLATION STUDY

**Compatibility with fast sampling solvers.** Our experiments use the default solver, DPMSolver++(Lu et al., 2022b), but we also demonstrate compatibility with more advanced solvers. As shown in Table 4, the performance improvements are consistent across different solvers. Notably, for all three solvers—EulerD (Karras et al., 2022), DEIS (Zhang & Chen, 2023), and DPMSolver++ (Lu et al., 2022b)—we observe significant gains, particularly in FID scores. EulerD shows a substantial improvement (FID: 39.688 → 35.735), as does DEIS (FID: 37.675 → 35.882).

**Effect of opposite denoising property alignment.** The $\Delta$-DiT framework uses $\Delta$-Cache for deep blocks during early sampling and for shallow blocks during later stages. In this experiment, we reverse the cache order, applying $\Delta$-Cache to shallow blocks in the early stages and deep blocks in the later stages. As shown in Table 5, for PIXART-$\alpha$, although the CLIP score shows a minor difference, FID and IS significantly deteriorate (FID: 35.882 → 41.374; IS also drops). While CLIP score reflects semantic alignment with text, FID and IS better capture the image's finer details, highlighting the effectiveness of the original caching strategy in enhancing image quality.

**Illustration of the increasing bound $b$.** Figure 6 illustrates the effect of the bound value on generation outcomes. As $b$ increases from 0 to 20, FID and IS improve and reach their optimal values around $b = 16$, while the CLIP score peaks at $b = 8$. Given that a decreasing CLIP score can significantly impact image-text alignment, we empirically determine that setting $b = 12$ offers the best trade-off between FID, IS, and CLIP score, balancing both image quality and semantic alignment.

**Illustration of the increasing number of cached blocks $N_c$.** Figure 7 depicts the effect of the number of cached blocks ($N_c$) on generation performance. As $N_c$ increases from 0 to 28, FID reaches its best value around $N_c = 14$, while IS and CLIP score peak around $N_c = 21$. To balance performance and acceleration, we select $N_c = 21$, which results in over 37% MACs reduction, as shown in Table 1.

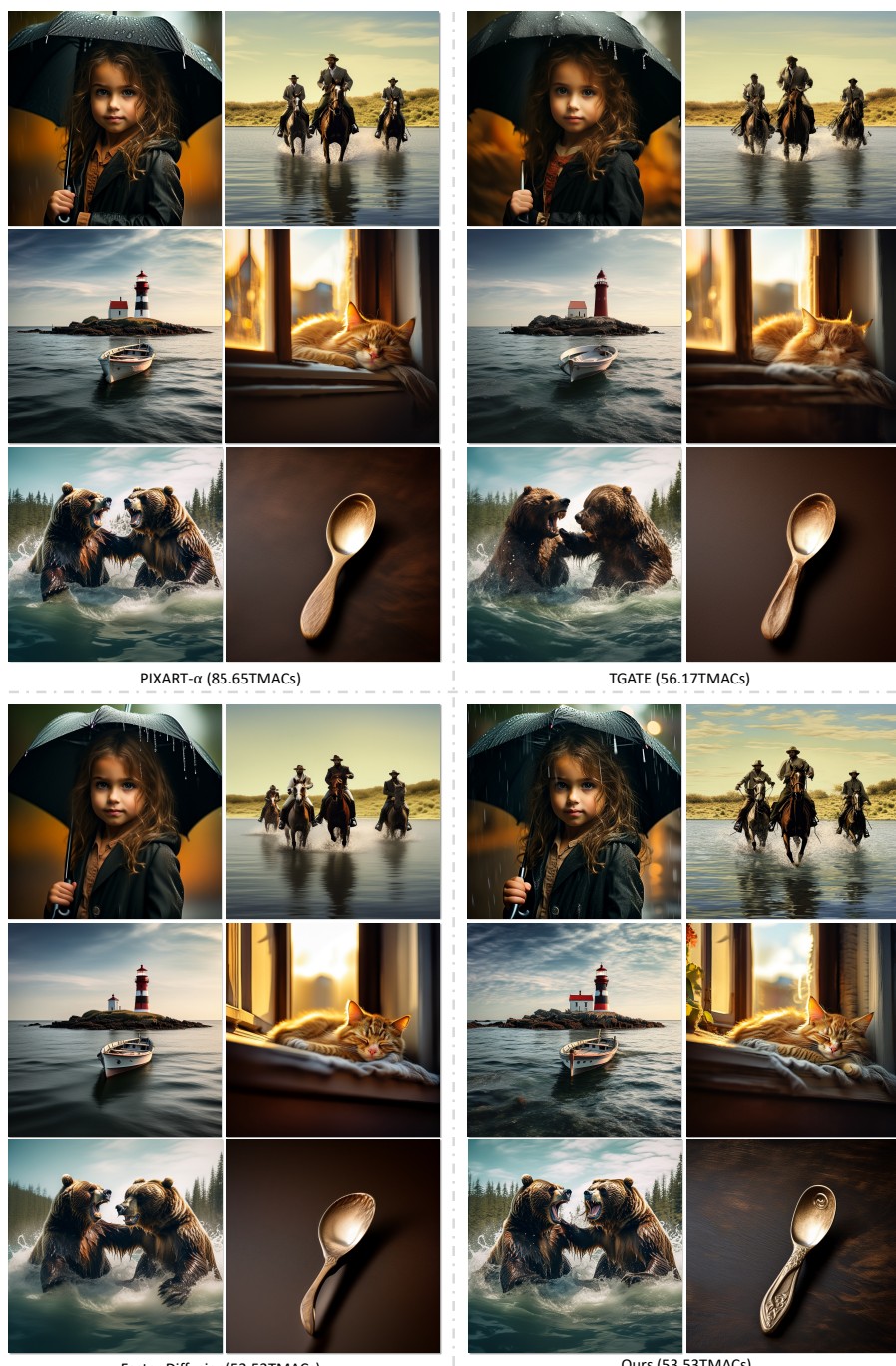

PIXART-α (85.65TMACs)    TGATE (56.17TMACs)

Faster Diffusion(53.53TMACs)    Ours (53.53TMACs)

Figure 8: **Comparison of images generated by various methods.** High-resolution images are generated based on different strategies using prompts randomly selected from six distinct scenes in the MS-COCO2017 dataset.

## 6 CONCLUSION AND LIMITATION

This paper considers the unique structure of DiT and proposes a training-free cache mechanism, $\Delta$-Cache, specifically designed for DiT. Furthermore, we qualitatively and quantitatively explore the relationship between shallow blocks in DiT and outline generation, as well as deep blocks and detail generation. Based on these findings and the denoising properties of diffusion, we propose the denoising property alignment acceleration method, $\Delta$-DiT, which applies $\Delta$-Cache to different part blocks of DiT at various denoising stages. Extensive experiments confirm the effectiveness of our approach. We believe that more refined search or learning strategies will yield even greater benefits.

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

# A ADDITIONAL METRIC EVALUATION

In the main text, we evaluate the performance of various methods using FID, CLIP, and IS metrics. Here, we have added more benchmarks for a more comprehensive evaluation. Specifically, we included CMMD (an upgraded version of the FID metric) (Jayasumana et al., 2024), as well as IR (ImageReward (Xu et al., 2024)) and HPSv2 (Wu et al., 2023b), two widely accepted human preference metrics. Refer to Table 6 for the specific experimental results. Our method obtains comprehensive optimal results on all those generation metrics.

Table 6: The MS-COCO2017 generation results for PIXART-$\alpha$ are evaluated. Gate is the hyper-parameter defined in TGATE (Zhang et al., 2024b). $T$ represents the number of timesteps, and $I$ indicates the starting block index for caching. Latency, measured in milliseconds, is tested on an Nvidia A100 GPU. Underscore and bold indicate the top 3 results.

| Method | MACs ↓ | Speedup ↑ | MS-COCO2017 | | | | | |
| | | | FID ↓ | IS ↑ | CLIP ↑ | CMMD ↓ | IR ↓ | HPSv2 ↑ |
|---|---|---|---|---|---|---|---|---|
| PIXART-$\alpha$ ($T = 20$) (Chen et al., 2023) | 85.651T | 1.00× | 39.002 | 31.385 | **30.417** | 1.104 | 3.961 | **29.466** |
| PIXART-$\alpha$ ($T = 13$) (Chen et al., 2023) | 55.673T | 1.54× | 39.989 | 30.822 | 30.399 | 1.113 | 4.265 | 29.338 |
| Faster Diffusion ($I = 21$) (Li et al., 2023b) | 53.532T | 1.60× | 42.763 | 30.316 | 30.227 | 1.119 | 5.048 | 29.009 |
| TGATE (Gate=8) (Zhang et al., 2024b) | 56.170T | 1.52× | 37.539 | 30.124 | 29.021 | 1.086 | 6.081 | 28.552 |
| Δ-Cache (Shallow Blocks) | 53.532T | 1.60× | 41.702 | 30.276 | 30.288 | 1.162 | 4.908 | 29.028 |
| Δ-Cache (Middle Blocks) | 53.532T | 1.60× | 35.907 | **33.063** | 30.183 | 1.091 | 4.160 | 29.229 |
| Δ-Cache (Deep Blocks) | 53.532T | 1.60× | **34.819** | 32.736 | 29.898 | **1.075** | 3.848 | 29.109 |
| Ours ($b = 12$) | 53.532T | 1.60× | 35.882 | 32.222 | 30.404 | 1.077 | **3.729** | 29.390 |

# B RESULTS ON STABLE DIFFUSION 3.0

In the main text, we mainly conducted experiments on the traditional classical DiT architecture (Peebles & Xie, 2023). Recently, some new DiT architectures have emerged, such as the MMDiT of SD3 (Esser et al., 2024). Therefore, we also evaluated the performance on these new DiT architectures, and the results are shown in Table 7. Even with the unique dual-branch architecture of SD3's DiT, our method remains applicable and achieves overall optimal performance in generation metrics, surpassing all baseline methods with comparable MACs.

Table 7: Results on the Stable Diffusion 3.0.

| Method | MACs ↓ | Speedup ↑ | MS-COCO2017 | | |
| | | | FID ↓ | IS ↑ | CLIP ↑ |
|---|---|---|---|---|---|
| SD3 ($T = 28$) (Esser et al., 2024) | 168.256T | 1.00× | 32.288 | **35.326** | **32.314** |
| SD3 ($T = 18$) (Esser et al., 2024) | 108.164T | 1.55× | 31.875 | 33.890 | 32.156 |
| Faster Diffusion ($I = 21$) (Li et al., 2023b) | 105.160T | 1.60× | 30.823 | 33.349 | 32.172 |
| Δ-Cache (Shallow Blocks) | 105.160T | 1.60× | 30.410 | 33.583 | 32.124 |
| Δ-Cache (Middle Blocks) | 105.160T | 1.60× | 30.595 | 33.902 | 32.065 |
| Δ-Cache (Deep Blocks) | 105.160T | 1.60× | 30.617 | 33.725 | 32.156 |
| Ours | 105.160T | 1.60× | **30.270** | 33.939 | 32.200 |

# C GENERATION PERFORMANCE UNDER DIFFERENT SPEEDUP

In the Table 1, we present the experimental results for a fixed speedup (1.6×). To demonstrate the performance of various methods under different speedups, we plotted the Pareto curve of CLIP-Score (more widely recognized in T2I tasks) versus computational cost, as shown in Figure 9. The red line represents the performance of our proposed method, which is positioned at the top-left of the performance curves of other methods, indicating overall Pareto-optimal results. And more results on LCM are shown in Table 8. Under the same speedup ratio (1.12×), our method achieves better generation results compared to existing methods. At a higher speedup ratio (1.4×), the proposed method still maintains an advantage in generation metrics, outperforming TGATE at a 1.12× speedup.

It is worth noting that Faster Diffusion fails to generate properly at a $1.12\times$ speedup, and TGATE's maximum speedup is only $1.17\times$. Our speedup ratio is groundbreaking, especially for challenging tasks like LCM.

Table 8: The MS-COCO 2017 results for the PIXART-$\alpha$-LCM model are evaluated, using the default number of generation steps, $T = 4$.

| Method | MACs ↓ | Speedup ↑ | Latency ↓ | MS-COCO2017 | | |
|---|---|---|---|---|---|---|
| | | | | FID ↓ | IS ↑ | CLIP ↑ |
| PIXART-$\alpha$-LCM (Chen et al., 2023) | 8.565T | 1.00× | 415.255 | 40.433 | 30.447 | 29.989 |
| Faster Diffusion ($I = 4$) (Li et al., 2023b) | 7.953T | 1.08× | 401.137 | 468.772 | 1.146 | -1.738 |
| Faster Diffusion ($I = 6$) (Li et al., 2023b) | 7.647T | 1.12× | 391.081 | 468.471 | 1.146 | -1.746 |
| TGATE (Gate=2) (Zhang et al., 2024b) | 7.936T | 1.08× | 400.256 | 42.038 | 29.683 | 29.908 |
| TGATE (Gate=1) (Zhang et al., 2024b) | 7.623T | 1.12× | 398.124 | 44.198 | 27.865 | 29.074 |
| Ours ($b = 2, N_c = 4$) | 7.953T | 1.08× | 400.132 | 39.967 | 29.667 | 29.751 |
| Ours ($b = 2, N_c = 6$) | 7.647T | 1.12× | 393.469 | 40.118 | 29.177 | 29.332 |
| Ours ($b = 2, N_c = 11$) | 6.883T | 1.24× | 350.539 | 42.653 | 29.810 | 29.689 |
| Ours ($b = 2, N_c = 16$) | 6.118T | 1.40× | 306.334 | 44.043 | 29.303 | 29.268 |

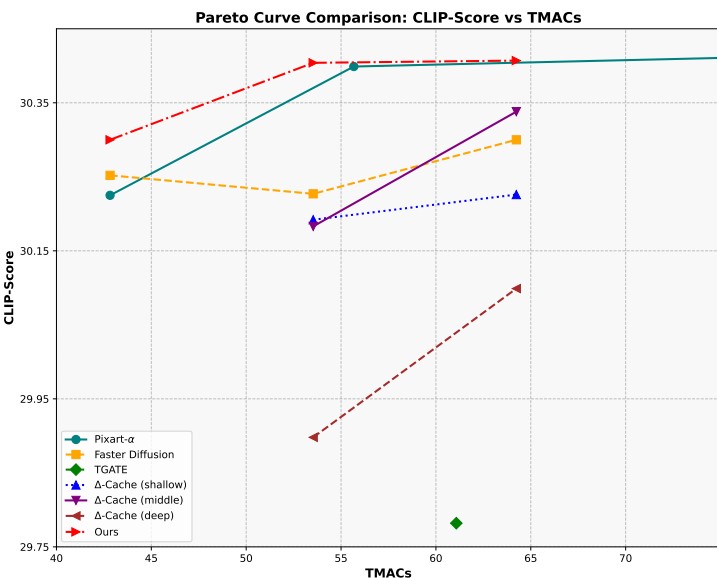

Figure 9: **Pareto curves of various methods' performance**: Evaluation results of the Pixart-$\alpha$ (Chen et al., 2023) on MS-COCO2017.

## D    Analysis of $\Delta$-Cache and $\Delta$-DiT

$\Delta$-Cache is the foundational module of $\Delta$-DiT, and $\Delta$-DiT utilizes alignment techniques built on top of $\Delta$-Cache. In this section, we visually demonstrate the advantages of $\Delta$-DIT over $\Delta$-Cache, which lacks alignment techniques. Figure 10 visualizes the generation results of different strategies. Similar to Section 4.1, strategies (b) and (c) have poor contour generation ability (an extra horse is generated), while (d) suffers significant detail loss (with many noise points in the image). On the other hand, strategy (e), which applies our alignment technique, maintains good overall contours and preserves details without introducing much noise.

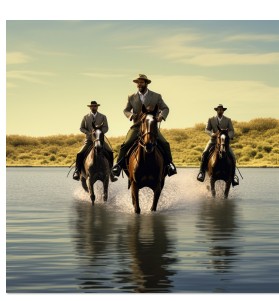
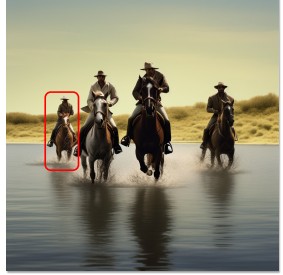
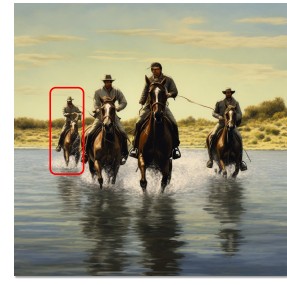

**(a) No Cache**

**(b) △- Cache the Shallow**

**(c) △- Cache the Middle**

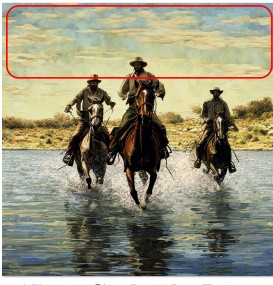
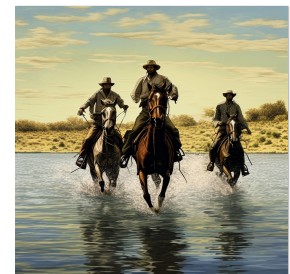

**(d) △- Cache the Deep**

**(e) △- DiT**

Figure 10: **Comparison of the images generated by our proposed methods.** The red line indicates areas with anomalies in the image. Compared to (a), both (b) and (c) show worse contour generation, with (b) and (c) introducing an extra horse in the contours. (d) exhibits poorer detail generation; while the contour is similar to (a), the image contains more noise. In contrast, (e) leverages alignment techniques, resulting in improved performance in both contour and detail generation.

## E ORTHOGONALITY WITH OTHER ACCELERATION METHODS

In this section, we demonstrate the orthogonality between $\Delta$-DiT and latent consistency models. In fact, our method can also be orthogonal to classical quantization methods. For the Pixart-$\alpha$, we performed INT8 quantization, and the resulting orthogonal outcomes are shown in Table 9. With the integration of quantization techniques, our speedup can reach $2\times$, while still maintaining good generation metrics.

Table 9: Orthogonal experimental results of our method and model quantization.

| Method | | | MS-COCO2017 | | |
|---|---|---|---|---|---|
| | **Latency** ↓ | **Speedup** ↑ | **FID** ↓ | **IS** ↑ | **CLIP** ↑ |
| Pixart-$\alpha$ (Chen et al., 2023) | 2290.668 | 1.00× | 39.002 | 31.385 | 30.417 |
| Pixart-$\alpha$ + Quantization | 1609.016 | 1.42× | 39.044 | 31.482 | 30.418 |
| Ours | 1534.551 | 1.60× | 35.882 | 32.222 | 30.403 |
| Ours + Quantization | 1114.004 | 2.06× | 35.855 | 32.305 | 30.394 |

## F EXPLORATION OF HYPERPARAMETER OPTIMIZATION METHODS

In $\Delta$-DiT, there are two hyperparameters: the boundary $b$ for detail generation and contour generation, and the number of cache blocks $N_c$. There are optimization techniques that can be applied to these hyperparameters. For example, by using FlashEval's fast evaluation algorithm to search for optimal values of $b$ and $N_c$. First, obtain 50 prompts that align well with CLIP metrics using the algorithm. Next, evaluate the CLIP score for different combinations of $b$ and $N_c$ using these prompts. Finally, identify the top 10 hyperparameter combinations, which provide the best text-image matching. Thanks to the speed of FlashEval's evaluation, this process takes about 6 GPU hours to run on a single A100 for the Pixart-$\alpha$. The quantitative results are shown in Table 10. It can be observed that

the CLIP score of the hyperparameters obtained through the search algorithm is better than that of the ones set by experience, further validating the effectiveness of the hyperparameter optimization algorithm.

Table 10: Results of searching for $N_c$ and $b$ after applying CLIP Score metric evaluation based on FlashEval (Zhao et al., 2024) method.

| Method | | MS-COCO2017 | | | |
|---|---|---|---|---|---|
| | $(N_c, b)$ | TMACs ↓ | CLIP ↑ | FID ↓ | IS ↑ |
| Pixart-$\alpha$ (Chen et al., 2023) | 0, None | 85.651 | 30.417 | 39.002 | 31.385 |
| Ours | 21, 12 | 53.532 | 30.403 | 35.882 | 32.222 |
| Ours + CLIP Search (Zhao et al., 2024) | 12, 10 | 67.297 | 30.472 | 37.547 | 31.409 |
| Ours + CLIP Search (Zhao et al., 2024) | 20, 8 | 55.061 | 30.445 | 38.670 | 31.330 |

## G  POTENTIAL WITHIN THE UNET ARCHITECTURE

Although $\Delta$-DiT is a method specifically designed for the DiT architecture, the $\Delta$-Cache concept we proposed is still applicable to the U-Net architecture. Specifically, the $\Delta$-Cache method can be applied to any position with the same resolution in U-Net, as shown in Figure 11. While the widely adopted DeepCache (Ma et al., 2023) uses feature maps as the cache target, our $\Delta$-Cache targets the difference in feature maps as the cache. Experiments on the SD1.5 Rombach et al. (2022) show that our method also achieves competitive results. For detailed quantitative data, refer to Table 11. Our method outperforms the DeepCache method across all three generation metrics under the same MACs.

Table 11: Applicability of our proposed $\Delta$-Cache on the U-Net architecture.

| Method | | MS-COCO2017 | | | |
|---|---|---|---|---|---|
| | MACs ↓ | Speedup ↑ | FID ↓ | IS ↑ | CLIP ↑ |
| Stable Diffusion v1.5 (Rombach et al., 2022) | 13.553T | 1.00× | 25.133 | **33.406** | 29.953 |
| DeepCache (Ma et al., 2023) | 7.923T | 1.71× | 23.313 | 32.620 | 30.146 |
| Ours ($\Delta$-Cache) | 7.923T | 1.71× | **23.117** | 33.014 | **30.148** |

## H  MORE FINE-GRAINED BLOCK ANALYSIS

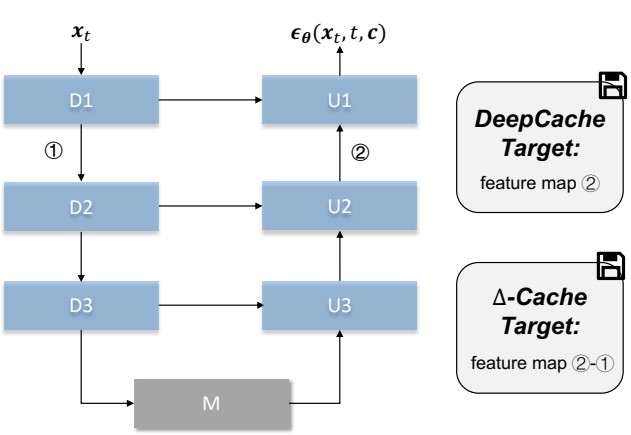

Figure 11: Comparison of $\Delta$-Cache and DeepCache in U-Net.

In the main manuscript, we explored the effect of blocks in three sections: shallow blocks (1-21), middle blocks (4-24), and deep blocks (8-28). Here, we present qualitative and quantitative results in a more fine-grained manner. Specifically, we applied $\Delta$-Cache to blocks 1-7, 7-14, 14-21, and 21-28, and the resulting qualitative and quantitative outcomes are shown in Figure 12. Qualitatively, we observed that applying $\Delta$-Cache to blocks closer to the shallow significantly impacts the contours compared to not using caching. For example, as shown in Figure 12b, a blue car is directly lost. In contrast, applying $\Delta$-Cache to later blocks has a more pronounced effect on the details. Quantitatively, when $\Delta$-Cache is applied to

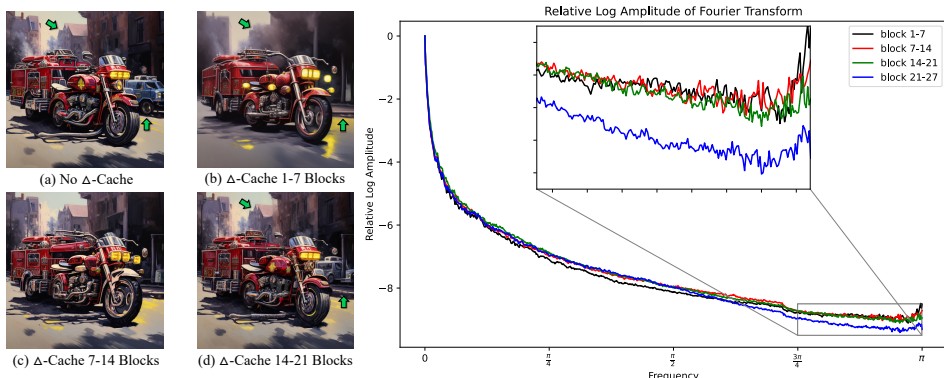

Figure 12: Qualitative and quantitative evaluation results of more fine-grained blocks division.

blocks 1-7, the loss of high-frequency information is minimal, while the loss of blocks 21-27 is large, which also means that the loss of detail is large. This conclusion aligns with the findings presented in the main manuscript.

## I    EXPERIMENTS IN MORE STEP SCENARIOS

Our goal is to generate images with an extremely small number of steps, so we set the generation process to 20 steps. Although the FID is slightly higher compared to results with more steps, it aligns with those reported in some literature under the same conditions. References (Zhang et al., 2024b) and (So et al., 2024) provide baseline results for 20-step PIXART-$\alpha$ and DiT-XL, respectively, with FID similar to ours. Finally, given that DiT-XL experiments are typically configured with 250 steps (cfg=1.5) (Peebles & Xie, 2023), we also conducted validation under this setting, as shown in Table 12. The experimental results are consistent with the findings in the paper, demonstrating that our method nearly outperforms the existing baseline approaches in terms of performance. Note that in our experiments, we used the pytorch-fid package to evaluate FID.

Table 12: Results on the DiT-XL (cfg=1.5). * indicates the results we replicated under the official code.

| Method | ImageNet-50k | | | |
| --- | --- | --- | --- | --- |
| | MACs $\downarrow$ | Latency $\downarrow$ | FID $\downarrow$ | IS $\uparrow$ |
| DiT-XL ($T = 250$) (Peebles & Xie, 2023) | 57.24T | - | 2.27 | 278.24 |
| *DiT-XL ($T = 250$) (Peebles & Xie, 2023) | 57.24T | 7064.60 | 2.29 | 277.67 |
| *DiT-XL ($T = 157$) (Peebles & Xie, 2023) | 35.94T | 4445.56 | 2.37 | 267.26 |
| Faster Diffusion ($I = 14$) (Li et al., 2023b) | 42.93T | 5338.08 | 2.63 | 261.40 |
| Faster Diffusion ($I = 21$) (Li et al., 2023b) | 35.77T | 4671.74 | 2.58 | 262.20 |
| $\Delta$-Cache (Shallow Blocks) | 35.77T | 4652.80 | 2.52 | 264.28 |
| $\Delta$-Cache (Middle Blocks) | 35.77T | 4641.43 | 2.37 | 270.84 |
| $\Delta$-Cache (Deep Blocks) | 35.77T | 4680.23 | 2.35 | 269.87 |
| Ours | 35.77T | 4642.53 | 2.31 | 271.03 |

