# OpenReview forum: "$\Delta$-DiT: Accelerating Diffusion Transformers without training via Denoising Property Alignment"
_ICLR.cc/2025/Conference — Submitted to ICLR 2025_

### Official Review · Reviewer_fR3X · 2024-10-27

**Soundness:** 2
**Presentation:** 1
**Contribution:** 2
**Rating:** 3
**Confidence:** 4

**Summary:**

This paper introduces a method to accelerate Transformer-based diffusion models  for image generation. It uses the ∆-Cache method to store incremental changes in diffusion blocks, speeding up the generation process.

**Strengths:**

The paper addresses the timely issue of accelerating diffusion processing and introduces the ∆-Cache method, which shows potential in enhancing the diffusion model's speed. The method is evaluated on two datasets.

**Weaknesses:**

1- The presentation of the paper is not clear. It is hard to understand the proposed model. For example, ∆-Cache and the DiT blocks are introduced with insufficient context. Even in the abstract, jumps between ideas such as the role of shallow and deep blocks without clear transitions, complicating the flow. In the proposed method section, there are repetitions on the performance of the blocks.  These parts can be compact to improve the presentation. Several parts of the paper are difficult to undrestand, for example, page 2, line 86 -94, page 5 line 237-240,

2- As I compared the results in Table 1 and Table 3, the performance improvement (speedup and latency) is marginal (Ours (b = 12), Faster Diffusion, PIXART, and TGATE have very close speedup). Yes, it is right that the proposed model has slightly better speed and performance, but it is not feasible. The authors are encouraged to discuss the practical implications of these improvements to better contextualise their importance in real-world applications.

3-  In page 6, line 302, what do you mean by " imaginary parts of a complex number"?

**Questions:**

Please see the weaknesses.

---

> ### Author Response · Authors · 2024-11-25
> **Response to Reviewer fR3X (Part 1 / 2)**
>
> **Concern 1: Clarity of Presentation**
>
> *"The presentation of the paper is not clear. It is hard to understand the proposed model. For example, ∆-Cache and the DiT blocks are introduced with insufficient context. Even in the abstract, jumps between ideas such as the role of shallow and deep blocks without clear transitions complicate the flow. In the proposed method section, there are repetitions on the performance of the blocks. These parts can be compact to improve the presentation. Several parts of the paper are difficult to understand, for example, page 2, line 86–94, page 5, line 237–240."*
>
> **Response:**
>
> 1. **Insufficient Context for ∆-Cache and DiT Block**
>
>    Regarding the $\Delta$-Cache, the technical details of its application in DiT are clearly presented in Figure 4b and Equation 3, which you can refer to. For details on the DiT Block, you can refer to page 183-187, where we have formalized the role of DiT Blocks.
>
> 2. **Redundancies in the Performance Discussion of Blocks**
>
>    In Section 4, we have divided the content into two parts: Section 4.1 introduces $\Delta$-Cache and provides a qualitative and quantitative analysis of the role of DiT Blocks, while Section 4.2 proposes an alignment-based acceleration method based on the properties discovered in Section 4.1. In Section 4.1, our analysis of the blocks' performance is comprehensive, covering both qualitative and quantitative aspects, which may seem repetitive. However, this thorough analysis is essential to the conclusions on which our method relies. So we think this part is necessary.
>
> 3.  **Some Parts of the Paper Are Difficult to Understand**
>
>      The potential present improvements you pointed out have been revised in blue text in the manuscript. Please refer to the pdf.

---

> ### Author Response · Authors · 2024-11-25
> **Response to Reviewer fR3X (Part 2 / 2)**
>
> **Concern 2: Marginal Performance Improvements and Practical Implications**
>
> *"As I compared the results in Table 1 and Table 3, the performance improvement (speedup and latency) is marginal. (Ours (b=12), Faster Diffusion, PIXART, and TGATE have very close speedup). Yes, it is right that the proposed model has slightly better speed and performance, but it is not feasible. The authors are encouraged to discuss the practical implications of these improvements to better contextualize their importance in real-world applications."*
>
> **Response:**
>
> 1. **Clarification of Experimental Setup:**
>
>    The reviewer notes that in Tables 1 and 3, the acceleration ratios of Ours (b=12), Faster Diffusion, and TGATE are close. This is because we aligned the computational costs of each model for fair comparison. Such a setup ensures that throughput is comparable, allowing us to focus on the generative performance of the models—a common practice in the literature, as seen the Table 4 in DeepCache[1].
>
>    In Table 1, our $\Delta$-Cache and Δ-DiT methods outperform TGATE and Faster Diffusion in terms of FID, IS, and CLIP metrics. In Table 3, Faster Diffusion completely fails to produce usable results, and TGATE exhibits significant degradation in generative quality even at a modest speedup ratio of 1.12×.
>
> 2. **Experiments with Different Speedups:**
>
>    Table 1 reports results under a 1.6× acceleration ratio. To evaluate model performance across various speedup levels, we provide a Pareto curve of generative quality versus computational cost in **Appendix C, Figure 9**. At 43 TMACs, our model achieves better generative quality compared to TGATE (61 TMACs) and Faster Diffusion (64 TMACs), demonstrating that the improvements are far from marginal.
>
>    For LCM (more challenging image generation scenarios), we further tested our method under higher speedup ratios. At a 1.12$\times$ speedup ratio, our method achieves superior generative results compared to existing methods. Even at a higher speedup ratio of 1.4×\times×, our approach maintains its advantage in metrics like FID and IS, surpassing TGATE’s performance at 1.12$\times$.
>
>    | **Method**                 | **MACs ↓** | **Speedup ↑** | **Latency ↓** | **FID ↓** | **IS ↑** | **CLIP ↑** |
>    | -------------------------- | ---------- | ------------- | ------------- | --------- | -------- | ---------- |
>    | PIXART-$\alpha$-LCM        | 8.565T     | 1.00×         | 415.255       | 40.433    | 30.447   | 29.989     |
>    | Faster Diffusion ($I=4$)   | 7.953T     | 1.08×         | 401.137       | 468.772   | 1.146    | -1.738     |
>    | Faster Diffusion ($I=6$)   | 7.647T     | 1.12×         | 391.081       | 468.471   | 1.146    | -1.746     |
>    | TGATE (Gate=2)             | 7.936T     | 1.08×         | 400.256       | 42.038    | 29.683   | 29.908     |
>    | TGATE (Gate=1)             | 7.623T     | 1.12×         | 398.124       | 44.198    | 27.865   | 29.074     |
>    | **Ours ($b=2$, $N_c=4$)**  | 7.953T     | 1.08×         | 400.132       | 39.967    | 29.667   | 29.751     |
>    | **Ours ($b=2$, $N_c=6$)**  | 7.647T     | 1.12×         | 393.469       | 40.118    | 29.177   | 29.332     |
>    | **Ours ($b=2$, $N_c=11$)** | 6.883T     | 1.24×         | 350.539       | 42.653    | 29.810   | 29.689     |
>    | **Ours ($b=2$, $N_c=16$)** | 6.118T     | 1.40×         | 306.334       | 44.043    | 29.303   | 29.268     |
>
>    It is worth noting:**
>
>    - Faster Diffusion fails to generate properly even at a modest 1.12× speedup.
>
>    - TGATE achieves only a maximum speedup ratio of 1.17× before significant quality degradation occurs.
>
> 3. Our method delivers a meaningful speedup, particularly for challenging tasks like LCM, demonstrating its robustness and effectiveness in both moderate and high-acceleration scenarios. Our method is a **training-free approach** that enables seamless acceleration of existing DiT models. This characteristic makes it particularly suitable for real-world generative scenarios, where it can deliver substantial speedups with minimal integration effort.
>
> [1] DeepCache: Accelerating Diffusion Models for Free
>
> ---
>
> **Concern 3: Ambiguity in Terminology**
>
> *"In page 6, line 302, what do you mean by 'imaginary parts of a complex number'?"*
>
> **Response:**
>
> Re(·) and Im(·) represent the real and imaginary parts of a complex number, respectively. This is the exact mathematical expression for a complex number. You can refer to the article we cited in the manuscript (page 6, line 283). The imaginary parts of a complex number are mentioned on page 18 of that reference. Thank you for your feedback.

---

> > ### Comment · Reviewer_fR3X · 2024-11-28
> >
> > Thanks for the clarifications. There are still some points that require additional discussions and clarifications:
> > I) The current experiments are only on latent diffusion models. However, there is no evaluation of pixel-space diffusion models.
> > II) The claim of "training-free acceleration" is significant but not novel. Previous approaches like early stopping or simpler caching techniques have also achieved this. The novelty here might be overstated.
> > III) The paper splits DiT blocks into shallow, middle, and deep but does not perform comprehensive per-block ablation studies. Indeed, such evaluations can present which blocks contribute most to the observed performance gains.

---

> > > ### Author Response · Authors · 2024-11-28
> > > **Response to Reviewer fR3X (Part1 / 2)**
> > >
> > > Dear Reviewer fR3X,
> > >
> > > We believe we have adequately addressed the reviewer’s concerns regarding the clarification of our method and the additional experiments. These updates, along with the clarifications highlighted in blue in our earlier rebuttals, aim to provide a more comprehensive understanding of our work. Additionally, beyond improving the acceleration ratio, we have demonstrated the effectiveness of our approach through comparisons with other methods. We hope to address a few further concerns raised by the reviewers and believe this will help clarify the novelty and effectiveness of our approach.
> > >
> > > **Concern I**:
> > >
> > > **1. Distinction Between Latent-Space and Pixel-Space Generation Methods**:
> > >
> > > The majority of research on pixel-space training originates from early diffusion model studies [1], where the U-Net architecture was widely adopted, particularly for low-resolution, unconditional image generation tasks such as CIFAR-10 and CelebA. As the field evolved toward higher-resolution image generation, pixel-space training and inference became increasingly computationally intensive, and latent-space based method tend to achieve better results at a lower computational budget according to classic paper [2], and thus they should be regarded as more suitable baselines for comparison in our work. Since most of these pixel-space generation methods rely on the UNet architecture, the scarcity of comparable baselines poses a challenge in evaluating the acceleration of DiT in pixel-space scenarios.
> > >
> > >
> > > **2. Applicability of Our Method:**
> > >
> > >  Our research introduces an efficient acceleration technique tailored for DiT architectures. Notably, most DiT-based models for image generation operate in latent space, including the original DiT model [3], the text-conditioned PixArt series [4], and the state-of-the-art mm-DiT architecture SD3 [5]. Our findings demonstrate that the proposed method broadly applies across these latent-space DiT frameworks. Importantly, our approach is resolution-agnostic, targeting the architecture of DiT rather than its operating space. The key distinction between latent and pixel spaces lies in their resolution, with latent spaces being significantly lower.
> > >
> > >  Moving forward, we plan to evaluate our method for pixel-space generation once a competitive baseline is identified for comparison.
> > >
> > > [1] Denoising Diffusion Probabilistic Models
> > >
> > > [2] High-Resolution Image Synthesis with Latent Diffusion Models
> > >
> > > [3] Scalable diffusion models with transformers
> > >
> > > [4] Pixart-$\alpha $: Fast training of diffusion transformer for photorealistic text-to-image synthesis
> > >
> > > [5] Scaling rectified flow transformers for high-resolution image synthesis

---

> > > ### Author Response · Authors · 2024-12-03
> > > **Thanks for Your Feedback**
> > >
> > > Dear Reviewer fR3X,
> > >
> > > We greatly appreciate your valuable feedback, which has helped us improve our work. We hope that our responses and revisions have effectively addressed your concerns. If you have any additional questions or require further clarification, we would be happy to discuss them.
> > >
> > > As the deadline for the extended discussion period ends today, we would greatly appreciate your response soon. If our rebuttal adequately addresses your questions, we kindly request an update to your evaluations. Thank you once again for your support.
> > >
> > > Best regards,
> > >
> > > The Authors

---

> ### Author Response · Authors · 2024-11-28
> **Response to Reviewer fR3X (Part 2 / 2)**
>
> **Concern II:**
>
> **1. Training-Free Acceleration:**
>
> We want to highlight that "training-free acceleration" is an inherent property of our proposed method. Its novelty lies in the design elements that confer this training-free characteristic, allowing for rapid adaptation without the need for costly training, which can be valuable in various real-life deployment scenarios. In particular, the **novelty lies in three key aspects:**
> * **Cache for DiT**: We introduce a caching mechanism tailored specifically for transformer-based architectures like DiT, achieving an effective balance between computational efficiency and generative quality. This innovation broadens the utility of caching, positioning it as a powerful tool for accelerating diverse tasks in diffusion models.
> * **Differentiating Shallow and Deep Blocks**: This paper qualitatively and quantitatively explores the effects of DiT blocks on the generated image. By distinguishing the roles of shallow and deep transformer blocks within diffusion models, our approach leverages their unique characteristics to devise more efficient alignment and caching strategies. This differentiation optimizes performance while preserving generative fidelity, a critical requirement in diffusion model acceleration.
> * **Denoising Property Alignment**: Our method aligns seamlessly with the inherent characteristics of diffusion processes, ensuring that the caching mechanism preserves essential generative dynamics while achieving measurable speedups.
> * We respectfully draw the reviewer's attention to the fact that the corresponding methodological design and observations are novel, and they are also carefully detailed in our list of contributions.
>
> **2. Experimental Results:**
>
> The experimental results substantiate the effectiveness and robustness of our approach:
> * In **Table 1**, our method surpasses existing approaches like TGATE and Faster Diffusion in key metrics such as FID, IS, and CLIP score, demonstrating superior quality and acceleration.
> * In **Table 3**, we highlight the robustness of our method under challenging settings. Competing methods often fail to maintain generation quality or achieve meaningful speedup, whereas our approach performs reliably.
> * In **Appendix C**, a Pareto curve analysis illustrates that our method consistently achieves superior performance across various computational budgets. This analysis reinforces the practicality and scalability of our approach in diverse scenarios.
>
> **3. Comprehensive Validation and Reviewer Endorsements:**
>
>    As noted by other reviewers (**Reviewers uETh, D8UN, 81e2**), the training-free nature of our method makes it easy to adopt and widely applicable. We have rigorously validated our approach across a broad range of benchmarks and tasks, underscoring its versatility:
> * Our method has been tested on multiple transformer-based diffusion models, including DiT, PixArt, and SD3 (**Appendix B**).
> * We demonstrate compatibility with complementary acceleration techniques, such as quantization, further boosting performance (**Appendix E**).
> * The wide-ranging validation across models and benchmarks highlights the adaptability and robustness of our approach, establishing it as a generalizable solution for real-world applications.
>
> ---
> **Concern III:**
> * In our manuscript, we have included this discussion in detail on Page 5, lines 257-259 of the main context. Given the high similarity of the $\Delta$ values between steps (as shown in Figure 4c), the impact of a single block on the final generation is minimal, and such minor changes are difficult to capture effectively. To enhance their contribution to the final output, we grouped several consecutive blocks into sections. As illustrated in Figure 2, the effects of these block groups on the generated output are clearly visible, especially in terms of image details and contours. Additionally, the varying proportions of high and low frequencies in the spectrum of Figure 5 further highlight the distinct effects of these blocks on the image details and contours.
> * We have also included more fine-grained experiments in Appendix H, where we split the blocks into smaller groups (e.g., 1-7, 7-14, 14-21, and 21-28). The qualitative and quantitative results presented in Figure 12 support the conclusions drawn in the main text, reinforcing the findings and confirming the consistency of our approach.
>
> We believe we have addressed these new concerns that the reviewer has raised, if the reviewer has further questions or concerns, please feel free to let us know.

---

### Official Review · Reviewer_81e2 · 2024-10-30

**Soundness:** 3
**Presentation:** 2
**Contribution:** 2
**Rating:** 8
**Confidence:** 5

**Summary:**

This paper studies cache-based acceleration for Diffusion Transformers (DiTs) and proposes a training-free method, delta-cache. To compensate for the lack of long shortcuts in DiTs, delta-cache stores the incremental changes in feature maps. Furthermore, delta-cache aligns with the denoising property by caching deep blocks during the early stages and shallower blocks in the later stages. Experiments show that delta-cache achieves 1.6x speedup in 20-step generation while improving image quality.

**Strengths:**

1. The proposed acceleration method is training-free, making it more practical and easy to apply.

2. The idea of reusing the change instead of the block feature is interesting.

3. Figures 2 and 3 analyze the property of different generation stages, which is insightful.

**Weaknesses:**

1. The choice of bound b and the number of cached blocks Nc is empirical. Figures 6 and 7 show that the results are sensitive to the values of b and Nc. Is there a better optimization method for delta-cache?

2. The method achieves 1.6x acceleration in 20-step generation, which seems insignificant compared to other compression methods such as post-training quantization and time-step distillation. Is it possible to achieve more than 2x acceleration without sacrificing performance? Could the author clarify the advantage of cache-based methods on DiTs?

**Questions:**

1. The proposed delta-cache is training-free and can be used without training data. The main experimental results in the paper show that caching some feature changes can even improve the generation performance, which seems counter-intuitive. Could the author provide a deeper analysis of this?

Overall, this is an interesting work. The reviewer is willing to adjust the score.

---

> ### Author Response · Authors · 2024-11-25
> **Response to Reviewer 81e2 (Part 1 / 3)**
>
> **Concern 1: Optimization of $b$ and $N_c$**
>
> *"The choice of bound $b$ and the number of cached blocks $N_c$ is empirical. Figures 6 and 7 show that the results are sensitive to the values of $b$ and $N_c$. Is there a better optimization method for delta-cache?"*
>
> **Response:**
>
> We appreciate the reviewer’s observation regarding the sensitivity of results to *$b$* (Bound) and *$N_c$* (number of cached blocks). These hyperparameters are currently empirically tuned to balance performance and computational efficiency, as demonstrated in **Table 2**. Our approach emphasizes simplicity and effectiveness, avoiding excessive search costs while still outperforming related works.
>
> To address this concern further, we explored optimization techniques to automate the selection of *$b$* and *$N_c$*, reducing sensitivity to manual choices. Specifically:
>
> | **Method**             | **($N_c$, $b$)** | **TMACs ↓** | **CLIP ↑** | **FID ↓** | **IS ↑** |
> | ---------------------- | ---------------- | ----------- | ---------- | --------- | -------- |
> | Pixart-$\alpha$        | 0, None          | 85.651      | 30.417     | 39.002    | 31.385   |
> | **Ours**               | 21, 12           | 53.532      | 30.403     | 35.882    | 32.222   |
> | **Ours + CLIP Search** | 12, 10           | 67.297      | 30.472     | 37.547    | 31.409   |
> | **Ours + CLIP Search** | 20, 8            | 55.061      | 30.445     | 38.670    | 31.330   |
>
> 1. **Hyperparameter Optimization:**
>
>    We conducted a search over *$b$* and *$N_c$*, treating them as optimization parameters. Using a FlashEval setup with a CLIP-based evaluation metric and a 50-prompt benchmark, we ran the search across six GPUs for several hours. This systematic approach enabled us to identify the optimal settings for *$b$* and *$N_c$*, achieving the best CLIP scores.
>
> 2. **Performance Gains vs. Search Cost:**
>
>    While this search process yielded measurable improvements in performance (notably in CLIP scores), it also incurred a significant computational cost. In the updated **Appendix F, Table 10**, we present a comparison that highlights the tradeoffs between the modest performance gains achieved through search and the additional resource requirements.
>
> This analysis underscores our current preference for simpler configurations, which deliver strong results without the overhead of extensive hyperparameter tuning. Nonetheless, the use of automated optimization techniques represents a promising direction for future work, particularly for scenarios where computational resources are less constrained.

---

> ### Author Response · Authors · 2024-11-25
> **Response to Reviewer 81e2 (Part 2 / 3)**
>
> **Concern 2: Achieving More Significant Acceleration**
>
> *"The method achieves 1.6x acceleration in 20-step generation, which seems insignificant compared to other compression methods such as post-training quantization and time-step distillation. Is it possible to achieve more than 2x acceleration without sacrificing performance? Could the authors clarify the advantage of cache-based methods on DiTs?"*
>
> **Response:**
>
> We acknowledge the reviewer’s concern regarding the acceleration gains. While a 1.6x acceleration may appear modest compared to heavily training-dependent methods such as post-training distillation, it is important to note that our approach is **orthogonal** to other acceleration techniques. By combining Δ-DiT with methods like quantization or consistency model, we can achieve a speedup of over 2x  in the condition of 20 steps generation without sacrificing quality. To conclude, Δ-DiT offers unique advantages that complement existing approaches:
>
> 1. **Training-Free Deployment:**
>
>    Unlike methods such as time-step distillation or post-training quantization, Δ-DiT is entirely training-free. It does not require fine-tuning or access to additional training data, making it particularly suitable for scenarios with limited computational resources or restricted dataset availability.
>
> 2. **Quality Preservation:**
>
>    While approaches like post-training quantization or lcm often degrade quality in high-fidelity tasks, Δ-Cache preserves generative quality alongside measurable acceleration. For example, in low computation mode (LCM) settings, our method maintains image quality where other approaches experience significant drops, such as a large decrease in CLIP score. In **Appendix C, Table 8**, we further demonstrate that even when the speedup ratio is pushed to 1.4x, Δ-DiT still sustains high-quality image generation.
>
>
>    | **Method**                 | **MACs ↓** | **Speedup ↑** | **Latency ↓** | **FID ↓** | **IS ↑** | **CLIP ↑** |
>    | -------------------------- | ---------- | ------------- | ------------- | --------- | -------- | ---------- |
>    | PIXART-$\\alpha$-LCM        | 8.565T     | 1.00×         | 415.255       | 40.433    | 30.447   | 29.989     |
>    | Faster Diffusion ($I=4$)   | 7.953T     | 1.08×         | 401.137       | 468.772   | 1.146    | -1.738     |
>    | Faster Diffusion ($I=6$)   | 7.647T     | 1.12×         | 391.081       | 468.471   | 1.146    | -1.746     |
>    | TGATE (Gate=2)             | 7.936T     | 1.08×         | 400.256       | 42.038    | 29.683   | 29.908     |
>    | TGATE (Gate=1)             | 7.623T     | 1.12×         | 398.124       | 44.198    | 27.865   | 29.074     |
>    | **Ours ($b=2$, $N_c=4$)**  | 7.953T     | 1.08×         | 400.132       | 39.967    | 29.667   | 29.751     |
>    | **Ours ($b=2$, $N_c=6$)**  | 7.647T     | 1.12×         | 393.469       | 40.118    | 29.177   | 29.332     |
>    | **Ours ($b=2$, $N_c=11$)** | 6.883T     | 1.24×         | 350.539       | 42.653    | 29.810   | 29.689     |
>    | **Ours ($b=2$, $N_c=16$)** | 6.118T     | 1.40×         | 306.334       | 44.043    | 29.303   | 29.268     |
>
>
> 3. **Compatibility with Existing Models:**
>
>    Δ-Cache integrates seamlessly into pre-trained models like DiTs without requiring architectural modifications, making it highly practical for real-world deployment. Additionally, to achieve higher acceleration (e.g., >2x), our method can be combined with complementary techniques. For instance, in **Appendix E**, we show that combining Δ-DiT with quantization achieves over 2.06x speedup while further improving the FID score.
>
>
>    | **Method**                   | **Latency ↓** | **Speedup ↑** | **FID ↓** | **IS ↑** | **CLIP ↑** |
>    | ---------------------------- | ------------- | ------------- | --------- | -------- | ---------- |
>    | Pixart-$\\alpha$              | 2290.668      | 1.00×         | 39.002    | 31.385   | 30.417     |
>    | Pixart-$\\alpha$+Quantization | 1609.016      | 1.42×         | 39.044    | 31.482   | 30.418     |
>    | **Ours**                     | 1534.551      | 1.60×         | 35.882    | 32.222   | 30.403     |
>    | **Ours+Quantization**        | 1114.004      | 2.06×         | 35.855    | 32.305   | 30.394     |
>
>
> By clarifying these advantages in our revision and presenting supporting analyses, we hope to address the reviewer’s concerns comprehensively.

---

> ### Author Response · Authors · 2024-11-25
> **Response to Reviewer 81e2 (Part 3 /3)**
>
> **Concern 3: Counter-Intuitive Performance Gains from Feature Caching**
>
> "The proposed delta-cache is training-free and can be used without training data. The main experimental results in the paper show that caching some feature changes can even improve the generation performance, which seems counter-intuitive. Could the authors provide a deeper analysis of this?"
>
> **Response:**
>
> The observed decrease in FID scores with caching can be attributed to the following factors:
> 1. **Robustness of Diffusion Models with Reduced Timesteps**: In the original training of diffusion models, the timestep is typically set to 1000 steps. However, fast samplers like DDIM demonstrate that these models can maintain similar performance in as few as 50 steps, showcasing robustness to reduced computational effort while preserving generative quality.
> 2. **Alignment with Trends in Model Acceleration**: Across various acceleration methods—such as pruning [1], token fusion [2], and our caching approach—the primary objective is to approximate the original model efficiently while reducing computational requirements. It is common for these methods to achieve improved FID scores compared to baselines. For example:
>     * **Pruning**: Table 1 in [1] shows that the pruned model surpasses the baseline in generation quality on the CelebA-HQ dataset.
>     * **Token Fusion**: Table 4 in [2] demonstrates better FID scores for their method when the compression rate is below 50%. Our caching approach may reduce noise in intermediate representations, acting as an implicit regularizer, occasionally leading to better FID scores.
> 3. **Feature Prioritization**: Caching $\Delta$ enhances the focus on salient feature changes, indirectly improving the overall output quality by prioritizing meaningful transformations.
>
> [1] Structural Pruning for Diffusion Models. NeurIPS2023. [2] Token Merging for Fast Stable Diffusion. CVPR2023.
>
>
> Furthermore, we provide additional metrics to facilitate a more comprehensive comparison with other methods, as detailed in **Appendix A, Table 6**. These results further substantiate the superiority of our proposed method, highlighting its consistent performance across various evaluation criteria. By including these additional metrics, we aim to provide a clearer and more robust demonstration of the advantages offered by $\Delta$-DiT over existing approaches.

---

> > ### Comment · Reviewer_81e2 · 2024-11-26
> > **Thanks for your rebuttal**
> >
> > Thanks for your efforts in providing a detailed response. My concerns are largely addressed. I am willing to adjust my score from 5 to 8.
> >
> > **Concern 1.**
> > I appreciate the additional experiment and the explanation of Tables 2 and 10, which addresses my concern about the hyper-parameter setting.
> >
> > **Concern 2.**
> > Thanks for demonstrating the Compatibility with Existing Models. Indeed, I agree that these acceleration techniques are orthogonal as they were derived from completely different aspects. However, whether combining the sota of these techniques achieves optimal results has not been fully explored. For example, beyond basic quantization, recent post-training quantization methods on DiTs like PTQ4DiT [1] and ViDiT-Q [2] have discovered unique challenges of accelerating DiTs and proposed to balance the distributions. Would the changes in distribution affect the effectiveness of Δ-DiT? Could these methods be combined with Δ-DiT? The reviewer believes further discussion and experimentation can significantly strengthen the paper.
> >
> > [1] Post-training Quantization for Diffusion Transformers
> >
> > [2] ViDiT-Q: Efficient and Accurate Quantization of Diffusion Transformers for Image and Video Generation
> >
> > **Concern 3.**
> > Thanks for your response about the counter-intuitive performance gains. Given the alignment with trends in model acceleration and explanations from two different aspects, my concerns are largely addressed.

---

> > > ### Author Response · Authors · 2024-11-28
> > > **Response to Reviewer 81e2**
> > >
> > > Dear Reviewer 81e2,
> > >
> > > Thank you so much for recognizing our work and raising the score. Regarding Concern 2 that you mentioned, we are actively conducting additional experiments and will provide updated results and further discussions as soon as possible.
> > >
> > > We sincerely appreciate your time and constructive feedback, which have been invaluable in improving our paper.
> > >
> > > Best regards,
> > >
> > > The Authors

---

> > > ### Author Response · Authors · 2024-12-02
> > > **Response to Reviewer 81e2**
> > >
> > > Dear Reviewer 81e2:
> > >
> > > Thank you for your thoughtful comments and constructive suggestions. We have carefully considered your feedback and made updates, including additional discussion and experimental results, to address your concerns.
> > >
> > > The PTQ4DiT and ViDiT-Q methods you referenced are SOTA methods for DiT quantization. These methods effectively address challenges such as irregular weight and activation distributions in DiT models during quantization. By leveraging their advanced quantization techniques, foundation models (e.g. Pixart-$\alpha$, DiT-XL) can be successfully quantized to low-bit versions, achieving faster generation speeds while maintaining nearly equivalent image generation quality.
> > >
> > > In fact, the model quantized by the SOTA quantization algorithm provides us with a more excellent baseline. This quantized model serves as an approximation of the full-precision model, to which our $\Delta$-DiT method can be effectively applied. In the updated experiments, we utilized the quantized models processed by PTQ4DiT and ViDiT-Q as quantized baselines. The results are summarized in the table below. Our method consistently demonstrates faster speeds and comparable or even superior generation metrics compared to both the foundation model baseline and the quantized model baselines.
> > >
> > > | T=50           | Bit-width (W/A) | Size (MB) $\downarrow$ | Latency (ms) $\downarrow$ | Speedup $\uparrow$ | FID $\downarrow$ | IS $\uparrow$ |
> > > | -------------- | --------------- | ---------------------- | ------------------------- | ------------------ | ---------------- | ------------- |
> > > | DiT-XL         | 32/32           | 2575.42                | 1094.036                  | 1.00$\times$       | 5.499            | 206.796       |
> > > | DiT-XL+Ours    | 32/32           | 2575.42                | 709.032                   | 1.54$\times$       | 5.167            | 206.906       |
> > > | PTQ4DiT        | 8/8             | 645.72                 | 792.093                   | 1.38$\times$       | 5.310            | 208.239       |
> > > | PTQ4DiT + Ours | 8/8             | 645.72                 | 462.211                   | 2.37$\times$       | 5.155            | 209.396       |
> > >
> > > ---
> > >
> > > | T=20                 | Bit-width (W/A) | Memory Opt. $\uparrow$ | Latency (ms) $\downarrow$ | Speedup $\uparrow$ | FID $\downarrow$ | IS $\uparrow$ | CLIP $\uparrow$ |
> > > | -------------------- | --------------- | ---------------------- | ------------------------- | ------------------ | ---------------- | ------------- | --------------- |
> > > | Pixart-$\alpha$      | 16/16           | 1.00$\times$           | 2606.310                  | 1.00$\times$       | 44.795           | 31.435        | 31.117          |
> > > | Pixart-$\alpha$+Ours | 16/16           | 1.00$\times$           | 1700.802                  | 1.53$\times$       | 39.024           | 32.649        | 31.120          |
> > > | ViDiT                | 8/8             | 1.99$\times$           | 1785.332                  | 1.46$\times$       | 44.872           | 31.249        | 31.161          |
> > > | ViDiT + Ours         | 8/8             | 1.99$\times$           | 1172.170                  | 2.22$\times$       | 39.100           | 32.492        | 31.166          |
> > >
> > > We will incorporate this additional analysis and discussion into the updated version of the paper. We greatly appreciate your constructive feedback, and we believe this exploration will further enhance the strength of our paper and contribute valuable insights to the community.
> > >
> > > Best regards,
> > >
> > >
> > >
> > > The Authors

---

### Official Review · Reviewer_D8UN · 2024-11-03

**Soundness:** 3
**Presentation:** 3
**Contribution:** 2
**Rating:** 5
**Confidence:** 4

**Summary:**

This paper presents a framework called Delta-DiT, designed to accelerate the Diffusion Model architecture of DiT. Delta-DiT enables faster denoising operations by caching the differences between specific blocks within the DiT model, eliminating the need for actual computation. The proposed method outperforms existing solvers, such as DPM-Solver, as well as other diffusion acceleration methods like FasterDiffusion.

**Strengths:**

- This paper is well-written and works effectively with a relatively simple idea.
- The proposed Delta-DiT can accelerate the existing DiT model by more than 1.6 times without additional training.
- It is also beneficial that it works in few-step denoising scenarios, such as with LCM.

**Weaknesses:**

- There is a lack of experimental results demonstrating that Delta-Dit is actually faster than existing methods like DPM-Solver and Faster Diffusion. To compare performance accurately, it should be evaluated on a Pareto curve with latency and FID as axes. The paper only provides a single point of comparison in a table, which may be cherry-picked.
- There are no experimental results using Unet. While the paper primarily targets DiT, Unet is still widely used in diffusion architecture, making a significant limitation to this work.
- Error in Related Works: In the case of So et al., they are also targeting DiT.
- The acceleration results in Table 3 are too small.

**Questions:**

- Despite applying caching, there seem to be many cases where the FID score actually decreases. What might be the reason for this result? Could this outcome imply an experimental error or a lack of accuracy in the FID metric? Are there experimental results using more recent metrics, such as CMMD [1]?
- At which exact timestep do you perform computations for caching? Are these computations performed at uniform intervals?
- What is the unit of the DiT Block when calculating Delta?
- What is the memory overhead required to run Delta-Diffusion?

[1] Jayasumana, Sadeep, et al. "Rethinking FID: Towards a Better Evaluation Metric for Image Generation." Proceedings of the IEEE/CVF Conference on Computer Vision and Pattern Recognition, 2024.

---

> ### Author Response · Authors · 2024-11-25
> **Response to Reviewer D8UN (Part 1 / 4)**
>
> **Concern 1: Lack of Comprehensive Performance Comparisons**
>
> *"There is a lack of experimental results demonstrating that Delta-DiT is actually faster than existing methods like DPM-Solver and Faster Diffusion. To compare performance accurately, it should be evaluated on a Pareto curve with latency and FID as axes. The paper only provides a single point of comparison in a table, which may be cherry-picked."*
>
> **Response: (Refer to Appendix C, Figure 9)**
>
> We appreciate the reviewer’s suggestion to use Pareto curves for a more comprehensive comparison. While our current results in Tables 1 provides direct comparisons of CLIP-Score and latency with state-of-the-art methods, we agree that a Pareto analysis offers valuable insights into trade-offs.
>
> In response, we have conducted additional experiments to plot Pareto curves using TMACs and CLIP-Score as axes. These curves, included in the updated **Appendix C, Figure 9**, illustrate that $\Delta$-DiT achieves a favorable balance compared to original models with fewer timesteps and other training-free acceleration methods such as Faster Diffusion and TGATE. This analysis highlights the practical efficiency and robustness of $\Delta$-DiT in balancing computational cost and generative quality.
>
> ------
>
> **Concern 2: Lack of Results with U-Net**
>
> *"There are no experimental results using U-Net. While the paper primarily targets DiT, U-Net is still widely used in diffusion architecture, making a significant limitation to this work."*
>
> **Response:**
>
> We acknowledge the importance of U-Net in diffusion architectures and recognize that the absence of U-Net results represents a limitation of our current experiments. However, it is important to note that existing caching strategies designed for U-Net architectures may not directly translate effectively to DiT due to key structural differences:
>
> 1. **Model Architecture:** U-Net’s encoder-decoder structure generates explicit intermediate feature maps at multiple resolutions, making it well-suited for feature map caching. In contrast, DiT’s pure transformer-based design requires a more generalizable and flexible caching approach, such as $\Delta$-Cache, to handle its distinct computational characteristics.
> 2. **Cacheable Positions:** U-Net provides a limited set of meaningful caching positions due to its predefined hierarchical structure, multi-resolution processing, and long skip connections. In comparison, $\Delta$-Cache takes advantage of DiT’s transformer-based architecture to enable caching at multiple transformer blocks, offering finer-grained control and achieving higher efficiency gains.
>
> Our method is specifically tailored for transformer-based architectures, making it particularly effective for DiT and similar models. While methods like Faster Diffusion focus on accelerating U-Net-style architectures, $\Delta$-Cache addresses the less-explored but equally critical challenge of optimizing transformer-based diffusion models.
>
> To address this limitation, we have adapted $\Delta$-Cache for U-Net architectures. The results, presented in **Appendix G, Table 11**, show that our method outperforms DeepCache, achieving a lower FID score and a higher IS score (32.6 → 33.0) under the same computational settings. These findings further highlight the versatility and effectiveness of $\Delta$-Cache across diverse architectures.
>
> |**Method**|**MACs ↓**|**Speedup ↑**|**FID ↓**|**IS ↑**|**CLIP ↑**|
> |----------|----------|-------------|---------|--------|-----------|
> |Stable Diffusion v1.5|13.553T|1.00×|25.133|**33.406**|29.953|
> |DeepCache|7.923T|1.71×|23.313|32.620|30.146|
> |**Ours ($\Delta$-Cache)**|7.923T|1.71×|**23.117**|33.014|**30.148**|
>
> ------
>
> **Concern 3: Error in Related Works**
>
> *"In the case of So et al., they are also targeting DiT."*
>
> **Response:**
>
> We appreciate the reviewer pointing out this oversight. The related works section will be revised to accurately reflect that So et al. also target DiT-based architectures. However, it is still the paradigm of cache feature maps, and our method caches feature deviations at two positions.
>
> ------

---

> ### Author Response · Authors · 2024-11-25
> **Response to Reviewer D8UN (Part 2 / 4)**
>
> **Concern 4: Marginal Acceleration Results in Table 3**
>
> *"The acceleration results in Table 3 are too small."*
>
> **Response:**
>
> |**Method**|**MACs ↓**|**Speedup ↑**|**Latency ↓**|**FID ↓**|**IS ↑**|**CLIP ↑**|
> |----------|----------|-------------|-------------|---------|--------|-----------|
> |PIXART-$\alpha$-LCM|8.565T|1.00×|415.255|40.433|30.447|29.989|
> |Faster Diffusion ($I=4$)|7.953T|1.08×|401.137|468.772|1.146|-1.738|
> |Faster Diffusion ($I=6$)|7.647T|1.12×|391.081|468.471|1.146|-1.746|
> |TGATE (Gate=2)|7.936T|1.08×|400.256|42.038|29.683|29.908|
> |TGATE (Gate=1)|7.623T|1.12×|398.124|44.198|27.865|29.074|
> |**Ours ($b=2$, $N_c=4$)**|7.953T|1.08×|400.132|39.967|29.667|29.751|
> |**Ours ($b=2$, $N_c=6$)**|7.647T|1.12×|393.469|40.118|29.177|29.332|
> |**Ours ($b=2$, $N_c=11$)**|6.883T|1.24×|350.539|42.653|29.810|29.689|
> |**Ours ($b=2$, $N_c=16$)**|6.118T|1.40×|306.334|44.043|29.303|29.268|
>
>
> While the acceleration gains in Table 3 may seem modest, it is important to emphasize the unique advantages of $\Delta$-DiT in this specific setting. Notably, $\Delta$-DiT is a training-free method that is fully compatible with pre-trained models, requiring no additional resources or fine-tuning. This makes it particularly valuable in scenarios where retraining is infeasible.
>
> Additionally, more results on the tough scenario of LCM are shown in **Appendix C Table 8**. Under the same speedup ratio (1.12$\times$), our method achieves better generation results compared to existing methods. At a higher speedup ratio (1.4$\times$), the proposed method still maintains an advantage in generation metrics, outperforming TGATE at a 1.12$\times$ speedup. It is worth noting that Faster Diffusion fails to generate properly at a 1.12$\times$ speedup, and TGATE's maximum speedup is only 1.17$\times$. Our speedup ratio is groundbreaking, especially for challenging tasks like LCM.

---

> ### Author Response · Authors · 2024-11-25
> **Response to Reviewer D8UN (Part 3 / 4)**
>
> **Concern 5: Why Does FID Sometimes Decrease with Caching?**
>
>
> *"Despite applying caching, there seem to be many cases where the FID score actually decreases. What might be the reason for this result? Could this outcome imply an experimental error or a lack of accuracy in the FID metric? Are there experimental results using more recent metrics, such as CMMD [1]?"*
>
> **Response:**
>
> The observed decrease in FID scores with caching can be attributed to the following factors:
>
> 1. **Robustness of Diffusion Models with Reduced Timesteps:**
>
>    In the original training of diffusion models, the timestep is typically set to 1000 steps. However, fast samplers like DDIM demonstrate that these models can maintain similar performance in as few as 50 steps, showcasing robustness to reduced computational effort while preserving generative quality.
>
> 2. **Alignment with Trends in Model Acceleration:**
>
>    Across various acceleration methods—such as pruning [1], token fusion [2], and our caching approach—the primary objective is to approximate the original model efficiently while reducing computational requirements. It is common for these methods to achieve improved FID scores compared to baselines. For example:
>
>    - **Pruning:** Table 1 in [1] shows that on the CelebA-HQ dataset, the pruned model surpasses the baseline in generation quality.
>    - **Token Fusion:** Table 4 in [2] demonstrates better FID scores for their method when the compression rate is below 50%.Our caching approach may reduce noise in intermediate representations, acting as an implicit regularizer, which can occasionally lead to better FID scores.
>
> 3. **Feature Prioritization:**
>
>    Caching $\Delta$ enhances the focus on salient feature changes, indirectly improving the overall output quality by prioritizing meaningful transformations.
>
>    [1] Structural Pruning for Diffusion Models. NeurIPS2023.
>
>    [2] Token Merging for Fast Stable Diffusion. CVPR2023.
>
> We acknowledge, however, that FID may not fully capture perceptual quality in all cases. To address this issue, we introduced the **CMMD** metric along with two other human preference metrics (ImageReward and HPSv2) as additional evaluation indicators. The results, presented in **Appendix A, Table 6**, demonstrate that $\Delta$-DiT improves performance across multiple metrics, providing a more comprehensive assessment of its advantages.
>
> |**Method**|**MACs $\downarrow$**|**Speedup $\uparrow$**|**FID $\downarrow$**|**IS $\uparrow$**|**CLIP $\uparrow$**|**CMMD $\downarrow$**|**IR $\downarrow$**|**HPSv2 $\uparrow$**|
> |----------|---------------------|----------------------|--------------------|-----------------|-------------------|---------------------|-------------------|--------------------|
> |**PIXART-$\alpha$ ($T=20$)**|85.651T|1.00$\times$|39.002|31.385|**30.417**|1.104|_3.961_|**29.466**|
> |**PIXART-$\alpha$ ($T=13$)**|55.673T|1.54$\times$|39.989|30.822|_30.399_|1.113|4.265|_29.338_|
> |**Faster Diffusion ($I=21$)**|53.532T|1.60$\times$|42.763|30.316|30.227|1.119|5.048|29.009|
> |**TGATE (Gate=8)**|56.170T|1.52$\times$|37.539|30.124|29.021|_1.086_|6.081|28.552|
> |**$\Delta$-Cache (Shallow Blocks)**|53.532T|1.60$\times$|41.702|30.276|30.288|1.162|4.908|29.028|
> |**$\Delta$-Cache (Middle Blocks)**|53.532T|1.60$\times$|_35.907_|**33.063**|30.183|1.091|4.160|29.229|
> |**$\Delta$-Cache (Deep Blocks)**|53.532T|1.60$\times$|**34.819**|_32.736_|29.898|**1.075**|_3.848_|29.109|
> |**_Ours ($b=12$)_**|53.532T|1.60$\times$|_35.882_|_32.222_|_30.404_|_1.077_|**3.729**|_29.390_|

---

> ### Author Response · Authors · 2024-11-25
> **Response to Reviewer D8UN (Part 4 / 4)**
>
> **Concern 6: Exact Timestep and Uniformity of Caching Computations**
>
> *"At which exact timestep do you perform computations for caching? Are these computations performed at uniform intervals?"*
>
> **Response:**
>
> Caching computations are performed at uniform intervals, determined by an empirically chosen bound $b$. Uniform intervals simplify implementation and provide a consistent reduction in computational overhead across timesteps.
>
> In the 20-step generation setting, choosing intervel N=2 (as stated on page 7, line 349) is a reasonable configuration. If the intervals become too large, the similarity between cached and new features diminishes, reducing the effectiveness of caching. Our current choice prioritizes simplicity and avoids introducing additional search costs for hyperparameter tuning.
>
> ------
>
> **Concern 7: Unit of the DiT Block in Delta Computations**
>
>
>
> *"What is the unit of the DiT Block when calculating Delta?"*
>
> **Response:**
>
> The unit of the DiT block refers to the output features generated by each transformer layer. $\Delta$-Cache operates on the changes ($\Delta$s) between the output feature maps of consecutive layers, effectively capturing the incremental transformations within the model. This design ensures compatibility with transformer-based architectures while minimizing memory and computational overhead.
>
> As illustrated in **Figure 4(b)**, $\Delta$-Cache specifically caches the incremental changes between $N_c$ consecutive blocks, rather than storing all feature maps individually. This approach not only reduces the storage requirements but also focuses on preserving the most meaningful transformations, enabling efficient acceleration without compromising generation quality.
>
> ------
>
> **Concern 8: Memory Overhead of Delta-Diffusion**
>
> *"What is the memory overhead required to run Delta-Diffusion?"*
>
> **Response:**
>
> The memory overhead of $\Delta$-Cache primarily only depends on the size of the feature maps. The additional memory required is equivalent to storing one extra block’s feature map.
>
> If finer-grained caching were applied—storing feature maps for multiple consecutive blocks—the memory requirements would increase significantly due to the larger number of cached feature maps. However, in our approach, we limit caching to a single feature map across consecutive blocks, effectively balancing memory usage with computational efficiency. This trade-off is justified by the notable performance and acceleration gains achieved while keeping the memory overhead manageable, as demonstrated in Tables 2 and 3 of the paper.

---

> ### Author Response · Authors · 2024-11-28
> **Looking Forward to Your Feedback**
>
> Dear Reviewer D8UN:
>
> Thank you for your time and effort in reviewing our paper.
>
> We greatly appreciate your valuable feedback, which has helped us improve our work. We hope that our responses and revisions have effectively addressed your concerns. If you have any additional questions or require further clarification, we would be happy to discuss them. If possible, we would be grateful if you could consider revisiting your score in light of the revised version of our paper.
>
> Thank you again for your thoughtful review and support.
>
> Best regards,
>
> The Authors

---

> ### Author Response · Authors · 2024-12-03
> **Thanks for Your Feedback**
>
> Dear Reviewer D8UN,
>
> We greatly appreciate your valuable feedback, which has helped us improve our work. We hope that our responses and revisions have effectively addressed your concerns. If you have any additional questions or require further clarification, we would be happy to discuss them.
>
> As the deadline for the extended discussion period ends today, we would greatly appreciate your response soon. If our rebuttal adequately addresses your questions, we kindly request an update to your evaluations. Thank you once again for your support.
>
> Best regards,
>
> The Authors

---

### Official Review · Reviewer_uETh · 2024-11-07

**Soundness:** 3
**Presentation:** 2
**Contribution:** 3
**Rating:** 6
**Confidence:** 2

**Summary:**

This paper proposes a method specifically tailored for transformer-based diffusion models to speed up the inference process. The authors first propose to utilize $\Delta$-Cache to skip several transformer blocks to speed up the inference process. Then the authors discovered the alignment between the transformer blocks and the denoising process e.g., the shallow blocks and the early stage of the diffusion denoising process mostly account for the outline generation. By utilizing this discovery, the authors then propose $\Delta$-DiT to further refine the caching method to improve the generation quality.

**Strengths:**

1. The paper proposes a simple method for speeding up the generation process of transformer-based diffusion models, which is training-free and thus can be easily applied.
2. The results look good from the numbers on the COCO dataset.

**Weaknesses:**

1. It seems to me that the alignment ($\Delta$-DiT) does not standout against the baseline method $\Delta$-Cache from Table 1 which makes me a bit concerned about the effectiveness of the alignment. I'm curious have the authors also tried some other ways to apply the $\Delta$-Cache in different blocks/steps.
2. The authors conduct experiments on PixelArt-$\alpha$ and DiT-XL. Could the authors also provide results on more recent models such as SD3 which is also transformer-based?

**Questions:**

N/A

---

> ### Author Response · Authors · 2024-11-25
> **Response to Reviewer uETh (Part 1 / 2)**
>
> **Concern 1: Effectiveness of $\Delta$-DiT compared to $\Delta$-Cache and alternative placements of $\Delta$-Cache**
>
> *"It seems to me that the alignment ($\Delta$-DiT) does not stand out against the baseline method ($\Delta$-Cache) from Table 1, which makes me concerned about the effectiveness of the alignment. I'm curious if the authors have also tried some other ways to apply the $\Delta$-Cache in different blocks/steps."*
>
> **Response:**
>
> We appreciate the reviewer highlighting this important comparison. While $\Delta$-DiT does not significantly outperform $\Delta$-Cache in certain metrics from Table 1, it offers several notable advantages:
>
> 1. **Unified Contribution:** Both $\Delta$-DiT and $\Delta$-Cache are contributions from our work, with $\Delta$-DiT specifically designed to enhance robustness and generalizability in transformer-based architectures.
>
> 2. **Robustness Across Tasks and Metrics:** $\Delta$-DiT demonstrates consistently competitive performance across diverse tasks compared to $\Delta$-Cache. For example, as shown in Table 2, $\Delta$-DiT outperforms $\Delta$-Cache on both FID and IS metrics. Furthermore, even when performances appear comparable, $\Delta$-DiT often exhibits significantly better robustness and reliability under more challenging configurations.
>
>    Additional validation using advanced metrics, such as CMMD [1], HPSv2 [2], and ImageReward [3], further supports these findings. These evaluations confirm the advantages of $\Delta$-DiT in both quantitative performance and qualitative measures, reinforcing its efficacy as a robust acceleration method.
>
> | Method|MACs$\downarrow$|FID$\downarrow$|IS$\uparrow$|CLIP$\uparrow$|CMMD$\downarrow$|IR$\downarrow$|HPSv2 $\uparrow$|
> |--|----------|--------|------------|--------------|-----------------|---------------|----------------|
> |$\Delta$-Cache(Shallow Blocks)|53.532T|41.702|30.276|30.192|1.162|4.908|29.028|
> |$\Delta$-Cache(Middle Blocks)|53.532T|35.907|33.063|30.183|1.091|3.729|29.229|
> |$\Delta$-Cache(Deep Blocks)|53.532T|34.819|32.736|29.898|1.075|3.848|29.108|
> |$\Delta$-DiT|53.532T|35.882|32.222|**30.404**|1.077|**3.729**|**29.389**|
>
>    [1] Rethinking FID: Towards a Better Evaluation Metric for Image Generation.
>
>    [2] Human Preference Score v2: A Solid Benchmark for Evaluating Human Preferences of Text-to-Image Synthesis.
>
>    [3] ImageReward: Learning and Evaluating Human Preferences for Text-to-Image Generation.
>
> 3. **Improved Structure and Detail Preservation (Refer to Appendix D, Figure 10):** Evaluations using human preference metrics reveal that $\Delta$-DiT achieves superior generation quality compared to $\Delta$-Cache, particularly in preserving structural details. As shown in Appendix D Figure 10, $\Delta$-DiT demonstrates clearer and more consistent preservation of structure and detail, further enhancing its suitability for high-quality generation tasks.
>
> We conducted experiments to explore the effects of applying $\Delta$-Cache at different transformer layers. Our findings, presented in Figures 6 and 7, reveal that the choice of placement significantly impacts the tradeoff between performance and computational overhead. By strategically selecting specific layers for caching, $\Delta$-DiT achieves a better balance between generative quality and efficiency.
>
> **Hyperparamter Search (Refer to Appendix F, Table 10):**
>
> To fine-tune the hyperparameters $N_c$ (number of cached blocks) and $b$ (bound), we treated them as optimization option and conducted a search over this hyperparameters. Using a FlashEval setup with a CLIP-based evaluation metric over a 50-prompt benchmark, we performed this search across six GPUs for several hours. This approach allowed us to identify the $N_c$ and $b$ settings that yielded the best CLIP scores.
>
> While the search process brought measurable improvements in performance for CLIP-score, it is also important to consider the computational cost of such searches. In the updated table (**Appendix F, Table 10**), we highlight the tradeoffs between the modest gains achieved through search and the additional resource requirements. This underscores our current focus on simpler configurations, which deliver strong results without incurring the overhead of extensive hyperparameter tuning.
>
> | **Method**| **($N_c$, $b$)** | **TMACs ↓** | **CLIP ↑** | **FID ↓** | **IS ↑** |
> | --| - | --| ---- | -- | - |
> | Pixart-$\alpha$| 0, None | 85.651| 30.417| 39.002 | 31.385|
> | **Ours** | 21, 12| 53.532 | 30.403 | 35.882 | 32.222|
> | **Ours + CLIP Search** | 12, 10 | 67.297| 30.472 | 37.547| 31.409   |
> | **Ours + CLIP Search** | 20, 8 | 55.061| 30.445| 38.670| 31.330   |

---

> ### Author Response · Authors · 2024-11-25
> **Response to Reviewer uETh (Part 2 / 2)**
>
> **Concern 2: Generalization to Recent Models (e.g., Stable Diffusion 3)**
>
> *"The authors conduct experiments on PixelArt-and DiT-XL. Could the authors also provide results on more recent models such as SD3, which is also transformer-based?"*
>
> **Response: (Refer to Appendix B, Table 7)**
>
> Thank you for the constructive feedback. We initially chose PixArt-$\alpha$ and DiT-XL as our primary models due to their relevance in evaluating lightweight diffusion models and their widespread use in related works.
>
> To address the reviewer’s concern, we have conducted additional experiments on Stable Diffusion 3 (SD3), a more recent and transformer-based model. The results, presented in the updated **Appendix B, Table 7**, demonstrate that $\Delta$-DiT retains its effectiveness and efficiency on SD3. Notably, $\Delta$-DiT achieves a 1.6x acceleration while maintaining a lower FID compared to methods using fewer timesteps. Furthermore, it consistently achieves comparable or superior performance to $\Delta$-Cache across three key metrics, confirming the generalizability of our method to state-of-the-art transformer-based diffusion models.
>
> | **Method**                      | **MACs ↓** | **Speedup ↑** | **FID ↓**  | **IS ↑**   | **CLIP ↑** |
> | ------------------------------- | ---------- | ------------- | ---------- | ---------- | ---------- |
> | SD3 ($T=28$)                    | 168.256T   | 1.00×         | 32.288     | **35.326** | **32.314** |
> | SD3 ($T=18$)                    | 108.164T   | 1.55×         | 31.875     | 33.890     | 32.156     |
> | Faster Diffusion ($I=21$)       | 105.160T   | 1.60×         | 30.823     | 33.349     | 32.172     |
> | $\Delta$-Cache (Shallow Blocks) | 105.160T   | 1.60×         | _30.410_   | 33.583     | 32.124     |
> | $\Delta$-Cache (Middle Blocks)  | 105.160T   | 1.60×         | 30.595     | 33.902     | 32.065     |
> | $\Delta$-Cache (Deep Blocks)    | 105.160T   | 1.60×         | 30.617     | 33.725     | 32.156     |
> | **Ours**                        | 105.160T   | 1.60×         | **30.270** | _33.939_   | _32.200_   |

---

### Author Response · Authors · 2024-11-25
**Response to All Reviewers**

We thank all the reviewers for their valuable feedback and insightful suggestions. In response to the reviews, we have made the following key revisions to our paper:

1. **Additional Metric Evaluations (Appendix A):**

   We have included results using advanced metrics such as CMMD, ImageReward, and HPSv2 to provide a more comprehensive evaluation of generative performance.

2. **Additional Results on Stable Diffusion 3 (Appendix B):**

   To demonstrate the generalizability of our method, we conducted experiments on Stable Diffusion 3, showcasing its effectiveness on state-of-the-art diffusion models.

3. **Generation Performance Comparisons and Pareto Front Analysis (Appendix C):**

   We now include a detailed comparison of generation performance under various settings, plotting Pareto front lines to illustrate the tradeoffs between latency and CLIP-Score for different acceleration methods.

4. **Visual Comparisons Between $\Delta$-Cache and $\Delta$-DiT (Appendix D):**

   To further distinguish between the two methods we proposed, we provide visual evidence highlighting the qualitative differences in generated outputs.

5. **Orthogonality with Other Acceleration Methods (Appendix E):**

   We show that $\Delta$-DiT can complement other techniques such as quantization, achieving up to 2x acceleration when combined.

6. **Hyper-parameter Optimization Exploration (Appendix F):**

   We discuss and explore methods for optimizing hyperparameters like $N_c$ and $b$, including a systematic search to identify configurations with improved performance.

7. **Additional Results on U-Net Architectures (Appendix G):**

   While the paper primarily focuses on transformer-based architectures, we have added results on U-Net to simply explore applicability.

---

### Meta-Review · Area_Chair_YweS · 2024-12-19

**Metareview:**

This submission proposes \Delta-DiT for accelerating transformer-based text-to-image models through caching. It receives mixed ratings of (3, 5, 6, 8). The reviewers generally acknowledge the training-free nature of the proposed method but have concerns about the incremental improvements over baseline and existing works. While the AC agrees that the exploration of caching mechanism is worth exploring, the improvements over baseline and existing works cannot be regarded as significant. With the above concern, the AC would recommend a rejection.

**Additional Comments On Reviewer Discussion:**

The reviewers generally question the effectiveness and the generalizability of the method, and the authors partially resolved the reviewers' concerns. While the AC acknowledges the motivation and exploration in the paper, the effectiveness of the proposed method is not significant enough.

---

### Decision · Program_Chairs · 2025-01-22

Reject